# *Streptomyces* umbrella toxin particles block hyphal growth of competing species

Qinqin Zhao[1], Savannah Bertolli[1], Young-Jun Park[2,3], Yongjun Tan[4], Kevin J. Cutler[1,5], Pooja Srinivas[1], Kyle L. Asfahl[1,6], Citlali Fonesca-García[7,8], Larry A. Gallagher[1], Yaqiao Li[1], Yaxi Wang[1], Devin Coleman-Derr[7,8], Frank DiMaio[3,9], Dapeng Zhang[4,10], S. Brook Peterson[1], David Veesler[2,3] & Joseph D. Mougous[1,2,6 ✉]

*Streptomyces* are a genus of ubiquitous soil bacteria from which the majority of clinically utilized antibiotics derive[1]. The production of these antibacterial molecules reflects the relentless competition *Streptomyces* engage in with other bacteria, including other *Streptomyces* species[1,2]. Here we show that in addition to small-molecule antibiotics, *Streptomyces* produce and secrete antibacterial protein complexes that feature a large, degenerate repeat-containing polymorphic toxin protein. A cryo-electron microscopy structure of these particles reveals an extended stalk topped by a ringed crown comprising the toxin repeats scaffolding five lectin-tipped spokes, which led us to name them umbrella particles. *Streptomyces coelicolor* encodes three umbrella particles with distinct toxin and lectin composition. Notably, supernatant containing these toxins specifically and potently inhibits the growth of select *Streptomyces* species from among a diverse collection of bacteria screened. For one target, *Streptomyces griseus*, inhibition relies on a single toxin and that intoxication manifests as rapid cessation of vegetative hyphal growth. Our data show that *Streptomyces* umbrella particles mediate competition among vegetative mycelia of related species, a function distinct from small-molecule antibiotics, which are produced at the onset of reproductive growth and act broadly[3,4]. Sequence analyses suggest that this role of umbrella particles extends beyond *Streptomyces*, as we identified umbrella loci in nearly 1,000 species across Actinobacteria.

Soil is typically home to a dense and diverse bacterial community, with many soils containing >10[9] bacterial species per gram[5]. Under such conditions, interference competition is intense, as evidenced by the wide range of interbacterial antagonism and defence systems that these bacteria harbour[6,7]. *Streptomyces* are a genus of ubiquitous soil bacteria that are notable for their production of antimicrobial secondary metabolites, many of which are used clinically as antibiotics[1,3,8]. Among other targets, *Streptomyces* spp. seem to use these antimicrobials to inhibit the growth of other *Streptomyces* spp., which suggests that interspecies antagonism within the genus is ecologically important[2]. In many bacteria, proteinaceous polymorphic toxins, in conjunction with their associated delivery machinery, mediate interspecies competition[9–16]. However, such systems have not yet been identified in *Streptomyces*.

Although polymorphic toxin delivery relies on distinct, sequence divergent machineries specific to the producer and target species, the small toxin domains they transport often share homology. A comprehensive bioinformatics study that exploited this feature to search for new polymorphic toxins found that the uncharacterized alanine leucine phenylalanine-rich (ALF) repeat proteins of *Streptomyces* and related organisms bear carboxy-terminal polymorphic toxin domains[15,17]. The

model streptomycete *S. coelicolor* encodes three ALF proteins, which we term umbrella toxin protein C1 (UmbC1), UmbC2 and UmbC3. Each contains an amino-terminal twin arginine translocation (TAT) signal, two sets of four ALF repeats (ALF1–ALF8), two extended coiled-coil domains, and variable C-terminal and toxin domains (Fig. 1a and Supplementary Tables 1 and 2). The ALF repeat is a degenerate (28% average identity across ALF1–ALF8 from UmbC1–UmbC3) 43–44 amino acid motif of unknown function[17] (Extended Data Fig. 1a).

## UmbC protein interaction partners

To initiate our investigation of the UmbC proteins, we modelled their conserved domains using AlphaFold[18]. The ALF repeat portion of the proteins consistently adopted a ring structure, with interactions between ALF1 and ALF5 closing the ring and ALF4 and ALF8 located opposite (Fig. 1b). The coiled-coiled domains of the proteins converged to form a stalk. In UmbC3, this stalk was predicted to extend unidirectionally the length of the domains, whereas the stalks of UmbC1 and UmbC2 adopted a bent configuration in initial models. Templating the models of UmbC1 and UmbC2 on UmbC3 using AlphaFold produced

[1]Department of Microbiology, University of Washington, Seattle, WA, USA. [2]Howard Hughes Medical Institute, University of Washington, Seattle, WA, USA. [3]Department of Biochemistry, University of Washington, Seattle, WA, USA. [4]Department of Biology, St Louis University, St Louis, MO, USA. [5]Department of Physics, University of Washington, Seattle, WA, USA. [6]Microbial Interactions and Microbiome Center, University of Washington, Seattle, WA, USA. [7]Plant Gene Expression Center, USDA-ARS, Albany, CA, USA. [8]Department of Plant and Microbial Biology, University of California Berkeley, Berkeley, CA, USA. [9]Institute for Protein Design, University of Washington, Seattle, WA, USA. [10]Program of Bioinformatic and Computational Biology, St Louis University, St Louis, MO, USA. ✉e-mail: mougous@uw.edu

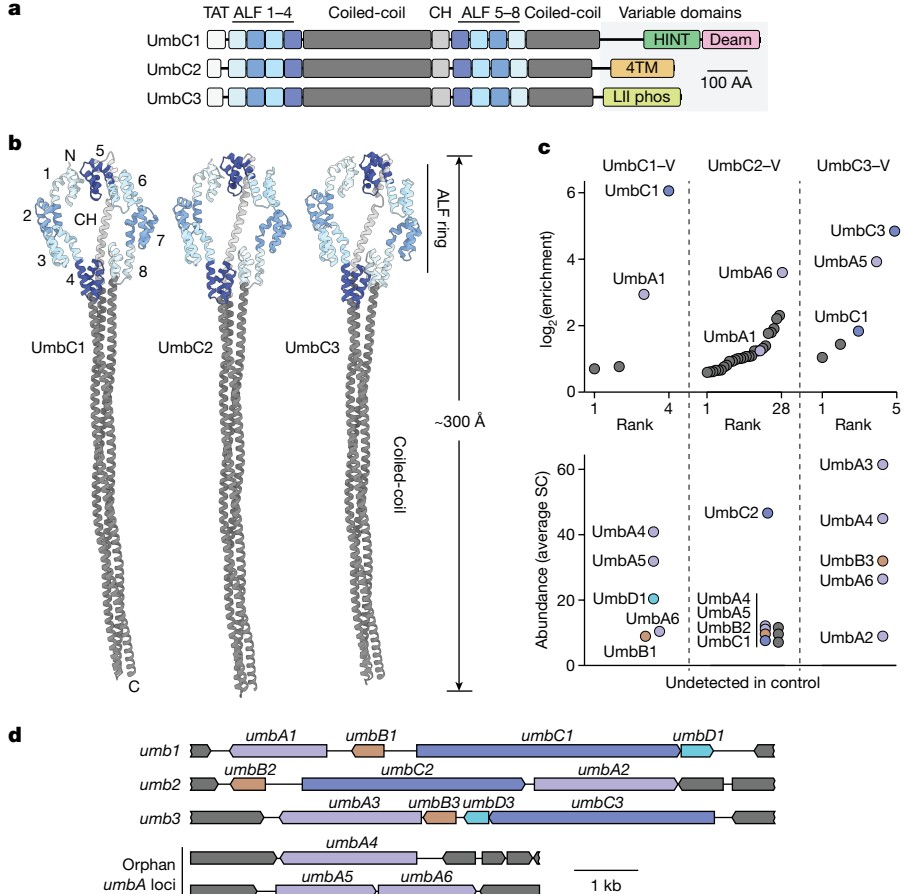

**Fig. 1 | *S. coelicolor* encodes three degenerate repeat-containing polymorphic toxins that interact with paralogous proteins. a**, Domain architecture of the UmbC proteins of *S. coelicolor*. Protein accession numbers and definitions of the variable C-terminal domains are available in Supplementary Tables 1 and 2. CH, connecting helix; Deam, deaminase; 4TM, 4TM tox; LII-phos, lipid II phosphatase, AA, amino acids. **b**, AlphaFold-predicted structural models of *S. coelicolor* UmbC proteins. UmbC1 and UmbC2 models were generated using template mode with UmbC3 as the reference. Colours correspond to **a**. ALF repeat numbering and location of the CH shown for UmbC1. The variable C-terminal domains, predicted to localize to the end of the stalk, could not be confidently modelled and are therefore not shown. **c**, IP–MS identification of proteins that interact with UmbC1, UmbC2 or UmbC3 from *S. coelicolor*. Top, average fold enrichment of proteins detected in both IP and control samples. Bottom, abundance (average spectral counts (SC)) for proteins detected only in IP samples. Colours indicate paralogous proteins; non-Umb proteins shown in grey. Note that additional background interacting proteins were identified for UmbC2, which we attribute to the lower abundance of this protein (46.5 SC) relative to UmbC1 (134.5 SC) and UmbC3 (781 SC) *n* = 2 biological replicates. V, VSV-G epitope. **d**, Loci encoding Umb protein complex components in *S. coelicolor*. Orphan *umbA* loci are those encoded distantly from other complex constituents. Colours consistent with **c**.

straight stalks for these proteins, a result consistent with the modelled structures we obtained by AlphaFold of several other UmbC proteins (Extended Data Fig. 1b). Overall, the proteins adopt a lollipop-like structure approximately 300 Å in length.

The UmbC structure we predicted is dissimilar to characterized proteins; therefore, it does not indicate how these proteins could function as polymorphic toxins. However, we reasoned that the ring arrangement of ALF repeats could serve as a platform for interaction with other proteins. To identify potential UmbC interaction partners, we generated *S. coelicolor* strains expressing C-terminally epitope-tagged UmbC1–UmbC3 from their native loci. Immunoprecipitation followed by mass spectrometry (IP–MS) analyses revealed candidate interaction partners for each UmbC protein (Fig. 1c and Supplementary Table 3). Sequence comparison of the proteins established two families, which we named UmbA and UmbB. We noted that each UmbC is encoded proximal to a *umbA* gene and the gene encoding the UmbB proteins it precipitates (UmbA1–UmbA3, UmbB1–UmbB3) (Fig. 1d). We also identified three UmbA proteins encoded outside these regions (UmbA4–UmbA6); these proteins co-precipitated with each UmbC protein. IP of UmbC1 also yielded an Imm1 immunity protein family member, which we named

UmbD1, as a candidate interaction partner. As observed for other polymorphic toxins, *umbD1* is located immediately downstream of its cognate toxin gene *umbC1*. We did not identify candidate immunity proteins for UmbC2 or UmbC3 in our data; however, a gene encoding an Imm88 immunity family protein (UmbD3) is located downstream of *umbC3*.

## Protein interactions in the Umb complex

The UmbA proteins of *S. coelicolor* consisted of a conserved N-terminal domain with high structural similarity to trypsin followed by a short helical linker to one (UmbA1–UmbA3, UmbA5 and UmbA6) or more (UmbA4) sequence divergent domains predicted to function as lectins (Fig. 2a, Extended Data Fig. 2a,b and Supplementary Tables 1 and 4). With the exception of an intervening additional lectin domain in UmbA4, these domains belonged to various β-propeller-fold lectin families[19]. Unlike the UmbA proteins, UmbB proteins did not share significant sequence or predicted structural relatedness to characterized proteins. The predicted structure of these small proteins consisted of an extended N-terminal disordered region linked by a short helix to a ten-stranded β-sandwich (Fig. 2b).

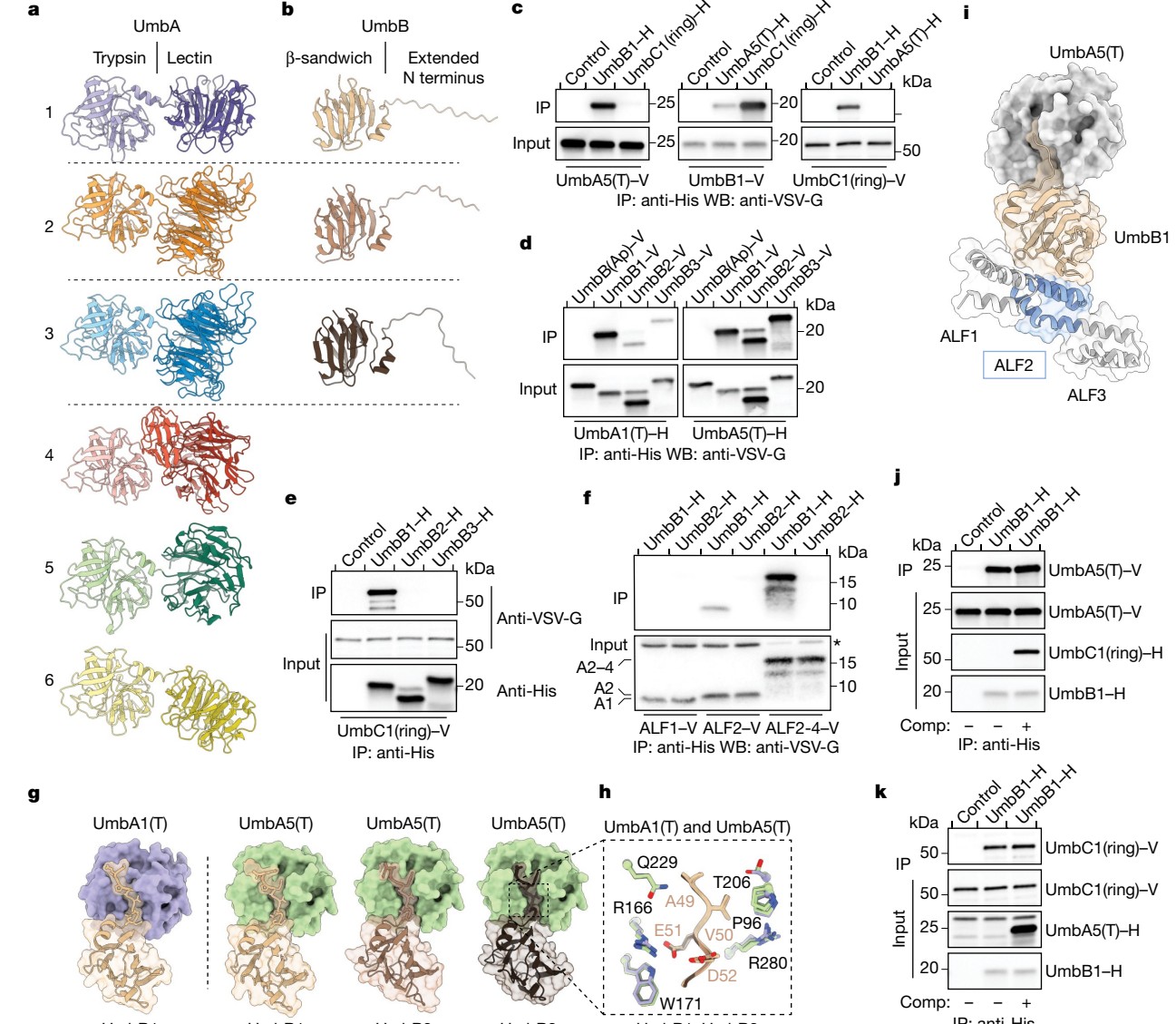

**Fig. 2 | PPIs in the Umb complex. a,b** Predicted structural models for UmbA1–UmbA6 (**a**) and UmbB1–UmbB3 (**b**) of *S. coelicolor*. Dashed lines separate pairs of proximally encoded proteins. **c–f**, Western blot (WB) analyses of IP experiments between the indicated heterologously expressed, tagged (hexahistidine (H) or VSV-G epitope (V)) Umb proteins. Control lanes correspond to beads in the absence of a bait protein. UmbB(Ap) is a UmbB protein from the distantly related species *A. philippinensis*. Bands corresponding to specific ALF repeats (ALF1 (A1), ALF2 (A2) and ALF2–ALF4 (A2–4)) and a background band (asterisk) are indicated in **f**. Additional input blots are provided in Extended Data Fig. 2. **g,h**, AlphaFold multimer-generated models for the interaction between the indicated UmbA and UmbB proteins of *S. coelicolor*, with surface representation highlighting the consistent predicted insertion of the N terminus of UmbB

proteins into the major cleft of UmbA trypsin domains. Additional predicted N-terminal disordered residues of UmbB1–UmbB3 are removed for clarity. Inset in **h** depicts strictly conserved residues in UmbA and UmbB in proximity to the modelled interaction interface. Side chains coloured as in **g**, and numbering corresponds to positions in UmbA5 and UmbB3. **i**, Ternary complex combining AlphaFold multimer models of UmbB1–UmbA5(T) and UmbB1–ALF2 of UmbC1. Flanking ALF repeats in UmbC1 (grey) are shown for context. **j,k**, WB analyses of competitive binding experiments between UmbB1 and its partners UmbA5(T) and UmbC1(ring). Purified competitor (Comp) UmbC1(ring)–H (**j**) or UmbA5(T)–H (**k**) were added in excess to IP experiments involving UmbB1 and UmbA5(T) or UmbC1(ring), respectively. Uncropped blots are provided in Supplementary Fig. 1.

Next we sought to interrogate protein–protein interactions (PPIs) between predicted Umb complex components. In these experiments, we focused on the trypsin domains of UmbA proteins (UmbA(T)) given the probable involvement of their C-terminal lectin domains in carbohydrate binding and the challenges we encountered trying to express their full-length form. Based on the assumption that PPIs involving UmbC would localize to the ALF repeats, we generated a DNA construct that fused the two sets of four repeats of UmbC1 into a ring, which removed the coiled-coil and C-terminal domains (UmbC1(ring)) (Extended Data Fig. 2c). Heterologous expression and co-IP studies provided evidence of direct interactions of UmbB1 with UmbA1(T), UmbA5(T)

and UmbC1(ring) (Fig. 2c,d and Extended Data Fig. 2d–f). Consistent with our *S. coelicolor* UmbC IP findings, UmbA1(T) co-precipitated more strongly with UmbB1 than with UmbB2 or UmbB3, whereas UmbA5(T) co-precipitated to a similar degree with UmbB1–UmbB3 (Fig. 2d). Neither UmbA1(T) nor UmbA5(T) co-precipitated with a UmbB from the distantly related organism *Actinoplanes philippinensis*.

In the UmbC ring, ALF1 and ALF5 were predicted to bind each other, apparently providing interactions important for uniting the two ring halves. Consequently, these repeats adopted an orientation and presented a solvent-accessible surface distinct from that of the other repeats (Extended Data Fig. 3). We reasoned that if ALF repeats mediate

UmbB binding to UmbC, this distinction would manifest as differential UmbB binding. Analyses of UmbB–UmbC interactions showed that UmbC1 displayed specificity for UmbB1, and that ALF2, but not ALF1, was sufficient to mediate this interaction (Fig. 2e,f). Furthermore, a construct composed of ALF2–ALF4 co-precipitated more efficiently with UmbB1 than the single ALF2 repeat, which indicated that multiple ALF repeats engage UmbB (Fig. 2f and Extended Data Fig. 2).

With experimentally determined PPIs between Umb proteins, we turned to AlphaFold to model their complexes. Notably, despite the sequence divergence among UmbB1–UmbB3 (39% average identity) and the trypsin domains of UmbA1 and UmbA5 (43% identity), the models consistently placed the extended N-terminal strands of UmbB1–UmbB3 into the prominent cleft of interacting UmbA proteins (Fig. 2g). In this configuration, a consensus tetrapeptide motif within the UmbB proteins (Ala-Val-Glu-Asp) contacts conserved UmbA residues lining their prominent groove, which corresponded to the substrate-binding cleft of trypsin proteins (Fig. 2h). One particularly strong predicted contact was a salt bridge between the Glu residue within this motif and Arg156 or Arg166 of UmbA1 or UmbA5, respectively. Non-conservative substitutions at these positions in UmbB1 and UmbA5 abrogated their interaction (Extended Data Fig. 4a). Despite the small size of UmbB, modelling suggested that the surfaces of UmbB1 that mediate UmbA and UmbC1 (ALF2) binding are non-overlapping (Fig. 2i). This idea was supported by our finding that excess UmbA5 or UmbC1 did not interfere with UmbC1 or UmbA5 binding to UmbB1, respectively (Fig. 2j,k).

Trypsin proteases utilize a Ser-His-Asp catalytic triad[20]. Alignment of UmbA1–UmbA6 with representative trypsin proteins showed that although the proteins share considerable sequence homology, no UmbA from *S. coelicolor* possessed the complete catalytic triad (Extended Data Fig. 4b). Moreover, we failed to detect catalytic activity from the purified trypsin domains of UmbA1 or UmbA5 using a universal trypsin substrate (Extended Data Fig. 4c,d). These data suggest that UmbA proteins utilize the trypsin fold in a non-canonical fashion to bind, but not cleave, the extended N terminus of their partner UmbB. This mode of binding seems to permit promiscuity in UmbA–UmbB interactions and leave a significant surface area of UmbB available for interactions with its other binding partner UmbC.

## Structure of the Umb1 particle

The network of PPIs we uncovered between Umb proteins, combined with the multiplicity of ALF repeats in UmbC, suggested that the proteins could assemble into a large, multimeric particle. Relative to UmbC2 and UmbC3, UmbC1-based affinity purifications were more homogenous and high yielding; however, instability near the C-terminal tagging site motivated us to use the C terminus of UmbA1 as an alternative site for isolating the complex by affinity chromatography (Fig. 3a, Extended Data Fig. 5a and Supplementary Table 3). We first isolated UmbA1 from the supernatant of a *S. coelicolor* strain expressing a C-terminally octahistidine-tagged allele of the protein from its native chromosomal locus. Subsequent separation by size chromatography produced a complex composed predominantly of UmbA1, UmbA4–UmbA6, UmbB1 and UmbC1 (Extended Data Fig. 5b). Transmission electron microscopy (EM) of this negative-stained sample revealed that Umb1 particles adopt an umbrella-like morphology, which led us to name these as umbrella (Umb) toxin particles (Fig. 3b and Supplementary Fig. 2). The long, slender stalk of these particles extended about 300 Å, whereas their crown had a width of around 250 Å.

Using single-particle cryo-EM, we obtained a structure of the Umb1 complex at an overall resolution of 4.3 Å (Extended Data Fig. 6, Extended Data Table 1 and Supplementary Figs. 3 and 4). We subsequently improved the resolvability of the regions constituting each spoke using local refinement, which produced reconstructions reaching up to 4.0 Å resolution. This process supported model building and provided a blueprint of the interactions underlying particle assembly. Our maps

enabled unambiguous placement of UmbC1 into the Umb1 particle. Although the entirety of the UmbC1 ring was clearly resolved at the centre of the umbrella crown, the density gradually decreased in quality towards the distal portion of its stalk. The C-terminal toxin and HINT domains of UmbC1, which, based on our model, would localize to the tip of the stalk, were therefore also not resolved in our map (Fig. 3c,d and Extended Data Fig. 6c,d). We postulate that flexibility within these regions relative to the rest of the Umb1 particle contributed to our inability to resolve this portion of the particle in our maps.

In line with our finding that UmbB1 interacts with individual ALF repeats, we were able to confidently model UmbB1 protomers in complex with ALF repeats at the base of each spoke. To model the UmbA portion of the spokes, we considered several factors. Our biochemical data showed that Umb1 particles possess four distinct UmbA proteins and that UmbB1 interacts with these in a promiscuous manner. Therefore, Umb1 particles can theoretically assume 1,024 possible configurations, ranging in subunit diversity from five copies of a single UmbA protein to all four UmbA proteins present, with one of them contributing two copies. Owing to the conserved UmbA trypsin-like domain engagement by the N terminus of UmbB1, these configurations are anticipated to share similar overall structures. Given that our map is derived from >350,000 individual Umb1 particles, we assumed that the UmbA portion of each spoke is effectively an ensemble of the four UmbA proteins in a stoichiometry consistent with their representation in our sample. This assumption posed a specific challenge for generating a discrete Umb1 particle model. We therefore elected to model UmbA1 at each spoke position because of the following reasons: UmbA1 is the cognate UmbA for the Umb1 particle; our purification of Umb1 particles on the basis of epitope-tagged UmbA1 ensured that UmbA1 populates at least one spoke in each particle imaged; UmbA1 is typical of Umb1-associated UmbA proteins in that it contains a single lectin domain (unlike UmbA4); and UmbA1 is accommodated well within our maps at each spoke position.

Our structure provided insights into many facets of Umb toxin biology. The interaction of the UmbA trypsin-like domain with UmbB1 placed the lectin domains of UmbA proteins at the distal ends of the Umb1 spokes. At this location, the domains are readily accessible to ligands, an arrangement compatible with a role in target cell receptor engagement. The structure confirmed that UmbC ALF1 and ALF5 do not bind UmbB1. Notably, it also revealed that ALF6 is not bound by UmbB1, producing a particle with five spokes rather than six (Fig. 3c,d). Subsequent IP experiments could not detect UmbB1–ALF6 binding, a result consistent with our structure (Fig. 3e). We therefore inspected the UmbB1–ALF interface to identify the molecular basis of this selectivity. In spite of substantial variability in their sequences, ALF repeats bound UmbB1 at a stereotyped location, with residues in two of its short helical segments providing many key contacts (Fig. 3f). At several positions within this region that are highly conserved across each UmbB1-binding ALF repeat, ALF6 harboured dissimilar amino acids (Fig. 3f and Extended Data Fig. 7a). To test the hypothesis that the amino acids in ALF6 at these positions prevent UmbB1 binding, we generated an ALF2 variant bearing the ALF6 residue at position four of its ALF repeat (ALF2(I4Q)). IP experiments demonstrated that this substitution abolished ALF2 binding to UmbB1 (Fig. 3e). Together, these findings provide an explanation for the lack of a sixth spoke in the Umb1 particle.

Our ability to link UmbB binding by ALF repeats to a major ultrastructural feature of Umb particles prompted us to explore whether the five-spoke arrangement of Umb1 is likely to be representative of other Umb particles. Notably, RoseTTAFold generated confident models for UmbB1–ALF complexes that closely matched those in our structure for each of the UmbB1-binding ALFs, but not for ALF6 or the other non-UmbB1-binding repeats (Extended Data Fig. 7b). Given this congruence with our experimental data, we used RoseTTAFold to model analogous complexes between UmbB2, UmbB3 and the ALF repeats of their corresponding UmbC proteins. As found for the UmbC1 ALF repeats, only ALF2–ALF4, ALF7 and ALF8 of UmbC2 and UmbC3 were

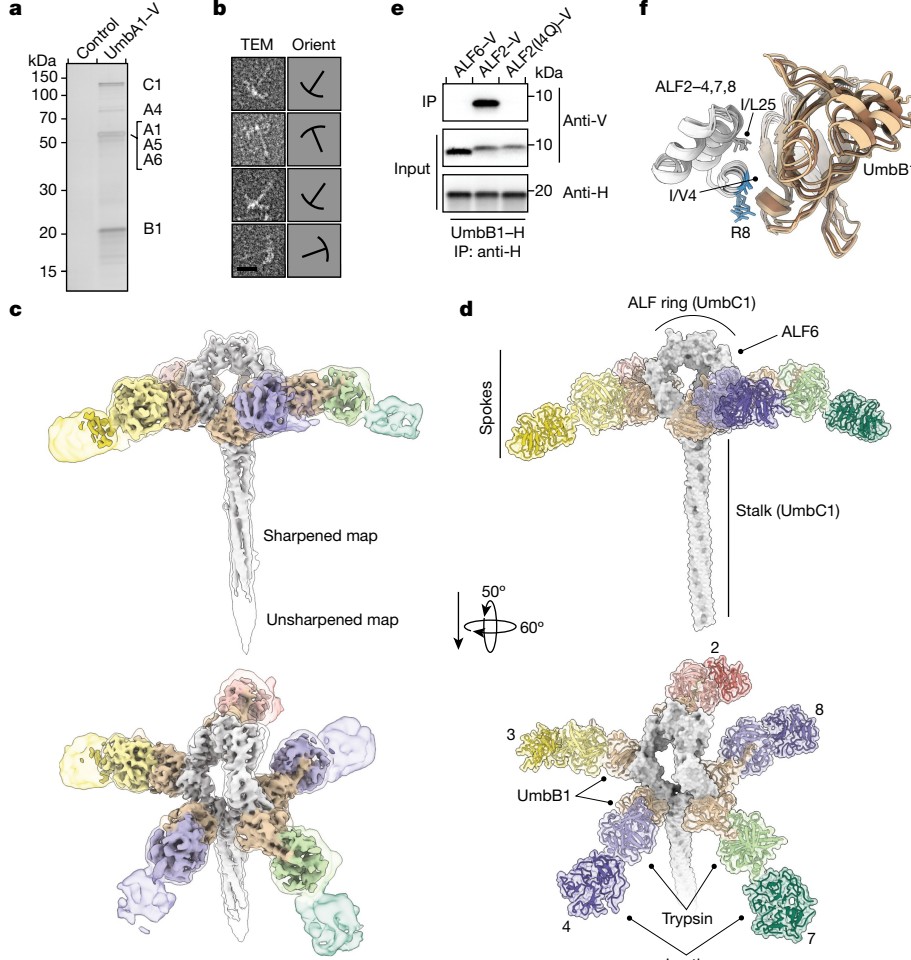

**Fig. 3 | Structure of the Umb1 particle. a**, Silver-stained SDS–PAGE analysis of the Umb1 protein complex precipitated from cultures of *S. coelicolor* with chromosomally encoded UmbA1–V. Control represents the equivalent precipitation from wild-type *S. coelicolor*. Experiment shown is representative of three independent replicates. **b**, Transmission EM analyses of negative-stained, purified Umb1 particles. Outlines indicating particle orientation (Orient) are shown on the right. Complete micrographs are provided in Supplementary Fig. 2a. **c**, Cryo-EM maps of the Umb1 particle at 4.3 Å unsharpened (transparent, 0.43 binarization threshold) and sharpened (opaque, 1.16). Maps are coloured based on model subunit proximity. Regions encompassing UmbB1 and UmbC1 are coloured as in Fig. 2, and regions encompassing UmbA1 are coloured differently in each spoke to highlight the complement of UmbA proteins that associate with the Umb1 particle. **d**, Depiction of the Umb1 particle. The depiction represents the model derived from our cryo-EM data, with the exception that the lectin domains present in spokes 2, 4 and 7 are shown but were not included in the deposited model owing to their weak density. The C-terminal domains of UmbC1 were not resolved in our structure and are not included in the depiction. Spoke numbers correspond to the interacting ALF repeat of UmbC1. ALF1, ALF5 and ALF6 (indicated) do not interact with UmbB1. **e**, WB analysis of IP experiments between UmbB1 and the indicated native and variant ALF repeats. Protein tags are indicated. **f**, Superimposition of UmbB1-binding ALF repeats (white) in complex measuring their corresponding UmbB1 protomer (brown shades) extracted from the Umb1 particle model. Conserved residues at the ALF–UmbB1 interface are shown, with several side chains truncated at Cβ as in the model. Those not conserved in ALF6 are coloured blue.

confidently predicted to associate with their respective UmbB proteins. This result indicated that the five-spoke configuration of the Umb1 particle may be a general feature of Umb toxins.

Finally, the Umb1 structure highlighted UmbB as a remarkable adaptor protein and keystone component of Umb toxin particles. That is, it interacts with five sequence divergent ALF repeats on one face and four different UmbA proteins on another. We are unaware of any other characterized protein that displays this degree of binding partner plasticity.

## A Umb toxin selectively targets streptomycetes

Functional predictions for the toxin domains associated with UmbC led us to speculate that Umb particles act on bacterial targets. Indeed, heterologous expression of the C-terminal domains of the UmbC proteins of *S. coelicolor* led to a significant reduction in bacterial viability (Fig. 4a). The toxin domain of UmbC1 was particularly potent in these

assays, and we confirmed the capacity of this predicted cytosine deaminase to introduce widespread C•G-to-T•A mutations in the DNA of intoxicated cells (Extended Data Fig. 8a–d). However, preliminary experiments measuring the impact of our purified Umb1 particle on the growth of a limited number of candidate bacteria did not identify clear targets of the toxin. To screen for Umb-targeted species in a more broad manner, we generated large quantities of concentrated Umb-particle-enriched supernatant (Umb supernatant) from cultures of wild-type *S. coelicolor* and a control strain bearing deletions in each *umb* locus (Δ*umb* supernatant) (Extended Data Fig. 9a). Next, we used this material to screen for toxin targets among a collection of 140 diverse bacteria. Given the propensity of polymorphic toxins to act on closely related organisms, we included an abundance of *Streptomyces* spp. and other actinobacterial species in our screen. This screen identified two candidate target species of the Umb toxin particles of *S. coelicolor* (*Z* score > 2.0), both of which are other *Streptomyces* spp.:

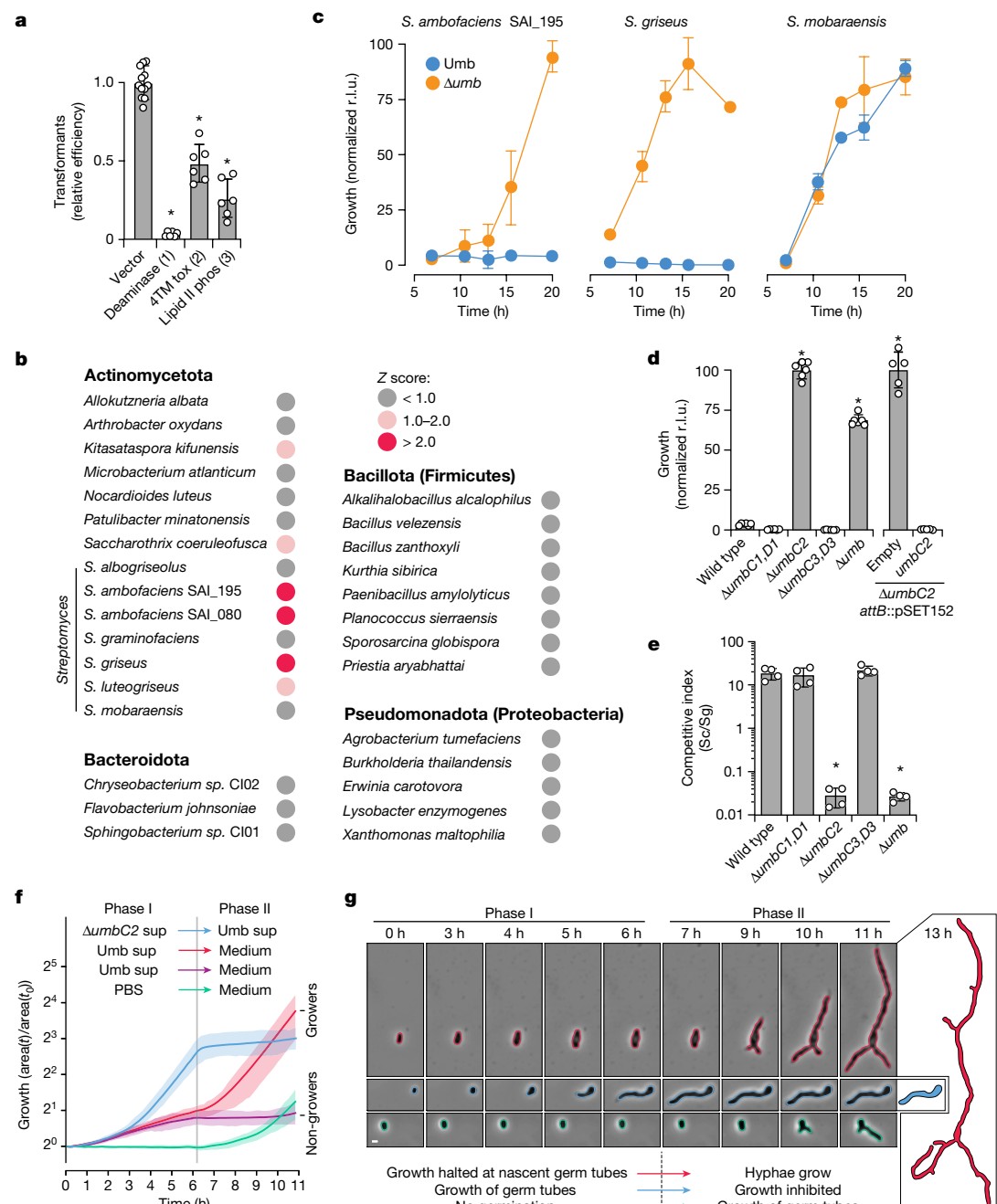

**Fig. 4 | Umb particles selectively inhibit vegetative hyphal growth of Streptomyces spp. a**, *Staphylococcus aureus* transformation efficiency of plasmids expressing UmbC toxin domains (1–3) relative to a vector control. Data represent the mean ± s.d. (*n* = 6 (experimental) or 12 (control)). **b**, Subset of Umb toxin susceptibility screening results. *Z* scores calculated from relative growth in the presence or absence of Umb toxins from two biological replicates of the screen; scores >2 indicate significant Umb-dependent inhibition. Additional strains screened shown in Extended Data Fig. 9b, and raw data provided in Supplementary Table 5. **c**, Growth of the indicated target and non-target strains treated with Umb or Δ*umb* supernatant (10% (v/v), measured as relative luminescence units (r.l.u.)). Colony-forming units quantified at 16 h available in Extended Data Fig. 9c. **d**, Growth yields of *S. griseus* after 16 h of treatment with *S. coelicolor* Umb supernatant from the indicated strains. **e**, Outcome of growth competition assays between the indicated strains of

*S. coelicolor* (Sc) and *S. griseus* (Sg). Data in **d** and **e** represent the mean ± s.d. (*n* = 3). **f**, Single-cell-based microscopy analysis of *S. griseus* growth as determined by cell area during exposure to the indicated treatments in a microfluidic flow cell. Phase I Umb supernatant-treated (sup) cells fell into two classes: those that resumed growth in toxin-free medium (growers) and those that did not (non-growers). Data for individual cells provided in Extended Data Fig. 9d and Supplementary Video 1. Shading indicates interquartile ranges. Red, *n* = 101; blue, *n* = 77; purple, *n* = 68; green, *n* = 78. **g**, Micrographs of representative cells from the indicated treatment groups in **f** outlined with Omnipose-generated segmentation masks. At 13 h, cell masks for two treatment groups are presented. Data shown in **f** and **g** are from one biological replicate representative of three. Scale bar, 2 μM. Asterisks indicate treatments significantly different from controls. *P* < 0.0001 (**a**,**d**) or *P* = 0.0003 (**e**), analysis of variance with Dunnett's multiple comparisons test, two-sided.

*S. ambofaciens* (three strains) and *S. griseus* (Fig. 4b, Extended Data Fig. 9b and Supplementary Table 5). Subsequent time-course experiments with these species and a control strain that was not a hit in our

screen demonstrated the capacity of *S. coelicolor* Umb supernatant to fully and specifically inhibit target cell growth in a manner dependent on Umb toxins. (Fig. 4c and Extended Data Fig. 9c).

The *S. griseus* strain hit in our screen is a type strain that is amenable to genetic manipulation and straightforward to cultivate[21]. We therefore selected this target organism to further characterize Umb-dependent toxicity. To identify the Umb particle (or particles) responsible for inhibiting *S. griseus* growth, we tested the toxicity of Umb supernatant derived from *S. coelicolor* strains unable to synthesize individual Umb particles. Inactivation of *umbC2*, but not *umbC1* or *umbC3*, abrogated the Umb supernatant growth inhibitory activity towards the organism (Fig. 4d). Genetic complementation of Δ*umbC2* further established the crucial role of UmbC2 in *S. griseus* growth inhibition by Umb supernatant. We next performed growth competition experiments to determine whether the level of Umb2 produced by *S. coelicolor* during co-culture is sufficient to intoxicate target cells. Notably, a *S. coelicolor* strain lacking Umb2 function was >600-fold less fit than the wild-type in co-culture with *S. griseus* (Fig. 4e). In summary, these data show that the secreted Umb toxins of *S. coelicolor* potently inhibit the growth of other *Streptomyces* spp.

## The Umb2 particle inhibits hyphal growth

Streptomycetes undergo a complex developmental programme that proceeds from spore germination to the formation of a hyphal network comprising a vegetative mycelium, followed by production of an aerial mycelium and sporulation. To gain insight into the possible ecological role of Umb toxin particles during competition among *Streptomyces*, we sought to determine the developmental stage at which target *Streptomyces* spp. are susceptible to Umb-particle-mediated intoxication. Single-cell-level analysis of time-lapse microscopy data revealed that Umb supernatant from wild-type *S. coelicolor* does not affect spore germination in the Umb2 target *S. griseus* (Fig. 4f,g, Extended Data Fig. 9d and Supplementary Video 1). Instead, similar to spores treated with culture medium or Δ*umbC2* supernatant, those treated with Umb supernatant increased in size and elaborated nascent germ tubes, phenomena not observed under conditions non-permissive to germination. However, spores treated with medium or Δ*umbC2* supernatant completed germination and formed hyphae, whereas Umb-supernatant-treated cells arrested at the nascent germ tube phase (Fig. 4f,g and Supplementary Video 1). Following replacement of the Umb supernatant with medium, a proportion of the population resumed vegetative growth after a variable lag period, whereas other cells remained inhibited (Extended Data Fig. 9d). We speculate that the vegetative bacterial surface area exposed to the Umb particle during germination determines the dose of toxin received, and therefore influences the subsequent fate of the cell.

Our data also revealed that the addition of Umb supernatant to actively growing mycelia produces an immediate, complete and persistent growth arrest (Fig. 4f,g and Supplementary Video 1). We did not observe lysis of intoxicated cells, a result consistent with the predicted pore-forming activity of UmbC. Together, these results demonstrate that the Umb2 particle acts specifically to inhibit the formation of vegetative mycelia in target organisms. Transcriptomics studies and our proteomics data showed that Umb toxins are also produced during this phase of the *Streptomyces* life cycle, which suggests that these toxins have a physiological function in mediating the outcome of competition among populations of vegetatively growing *Streptomyces*[22,23]. This effect is distinct from that induced by small-molecule antimicrobials produced by streptomycetes, which generally target a much broader group of organisms for the purpose of limiting access to nutrients released by lysed kin cells during aerial hyphae formation[3].

## Diversity and distribution of Umb toxins

The Umb particles of *S. coelicolor* confer a significant advantage in competition with at least two species. Given the prevalence of antagonistic interactions among bacterial species, we reasoned that others might

harbour and utilize Umb toxins in an analogous fashion. Leveraging our *S. coelicolor* findings pertaining to the particle constituents and genetic organization of Umb1–Umb3, we searched publicly available bacterial genomes to broadly define the distribution of Umb toxins. In total, we identified 1,117 genomes, deriving from 875 species, that we predicted to possess the capacity to synthesize one or more Umb particles (UmbB and UmbC within ten genes of each other) (Supplementary Table 1). More than half of these corresponded to species within the order Streptomycetaceae; the remaining *umb* loci-containing species were distributed among six other orders of Actinobacteria (Fig. 5a). In multiple bacteria capable of synthesizing distinct Umb particles, we identified UmbA proteins encoded at loci unlinked to those encoding UmbB and UmbC (Supplementary Table 1 and Supplementary Fig. 5). This result suggests that the association of 'orphan' UmbA proteins with multiple particles, as observed in *S. coelicolor*, may be common. It is notable that we did not find support for Umb particle production by bacteria outside Actinobacteria. If the action of Umb toxins is restricted to related species or to bacteria that exhibit mycelial growth, this finding could reflect the phylogenetic limits of targeting through this mechanism.

We found 77 divergent toxin families associated with the UmbC proteins identified in our analyses (Supplementary Table 2). Although many of these had sequence similarity to toxin domains associated with other polymorphic toxin systems, many, including the two most frequently observed in UmbC proteins, represented previously unrecognized families (4TM tox, Ntox71). Functional predictions suggested that as a group, Umb toxins act upon a marked range of essential cellular processes (Fig. 5b and Supplementary Table 2).

A distinct feature of Umb particles uncovered by our work in *S. coelicolor* is their incorporation of variable lectin domains through promiscuous UmbA binding. Taken together with their accessibility at the ends of Umb particle spokes, we propose that these domains mediate target cell binding and, at least in part, underpin the species selectivity of intoxication that we observed. Examination of the 882 UmbA proteins identified by our search highlighted extraordinary family-level and within family-level diversity in the lectin domains associated with these proteins (Supplementary Table 4). Moreover, we identified marked structural diversity among UmbA proteins, including those that, like *S. coelicolor* UmbA4, encode multiple distinct lectin domains, and others that are fused to UmbB-like domains (Fig. 5c). AlphaFold models of the latter predicted that despite their fusion, the predominant engagement mode of the two domains mirrors that which we identified for the individually encoded proteins. That is, an extended N-terminal structure of the UmbB domain inserts within the major cleft of the trypsin-like domain. Taken together, the diversification of toxin and lectin domains associated with Umb toxin particles provides evidence for a molecular arms race among producer and target cells, wherein target cells can escape intoxication either by receptor modification or by acquiring a downstream, direct toxin resistance mechanism.

## Discussion

Umb toxin particles represent a previously unrecognized component of the antibacterial arsenal of *Streptomyces*. We postulate that Umb particles mediate dynamic short-range antagonism among the vegetative mycelia of competing species vying for the same niche. This would provide the evolutionary pressure driving Umb particle selectivity and diversification, as the overlap in niches of highly related bacteria increases their probability of repeated encounters[2,6,24]. The chemical and biophysical properties of Umb particles are also consistent with this role. Umb toxin particle complexity and apparent vulnerability to proteases or other insults suggests that they are short-lived and therefore unable to act at longer length scales. Indeed, these properties of the Umb particles may underlie why such potent

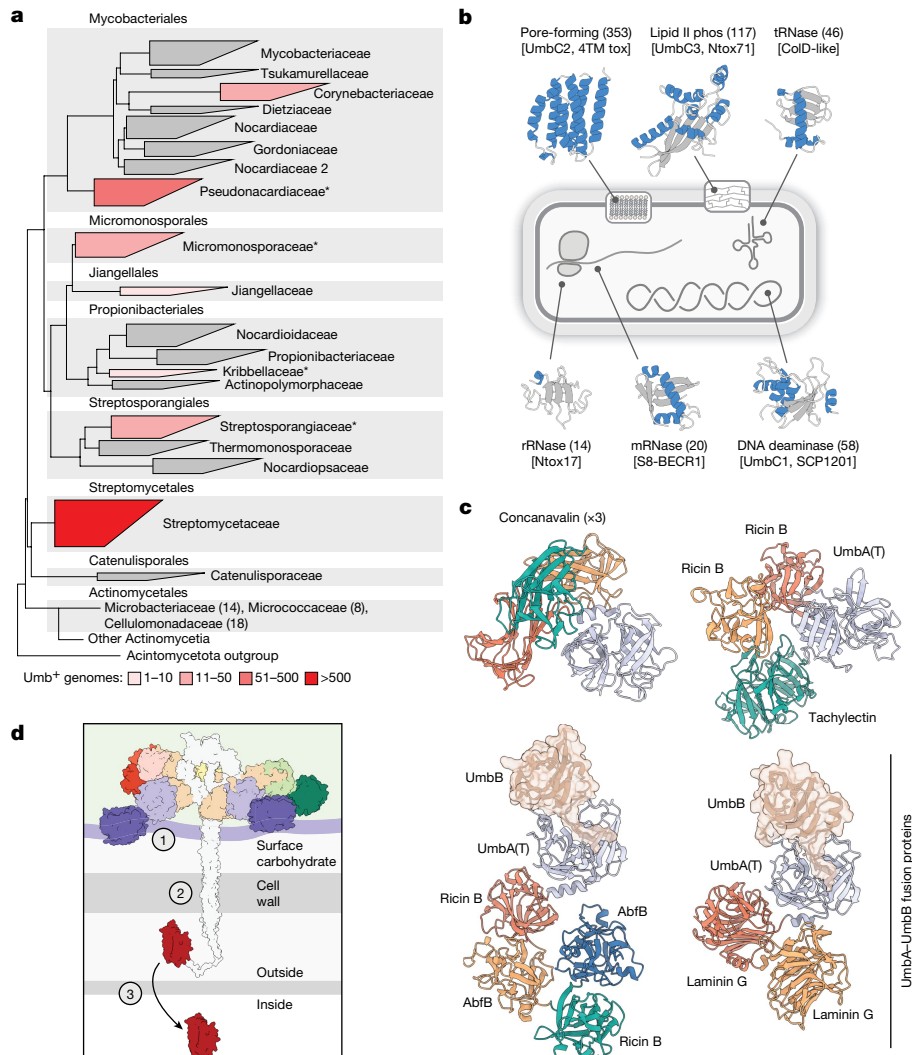

**Fig. 5 | Phylogenetic distribution and functional diversity of Umb proteins.**
**a**, Phylogenetic tree of orders and families within Actinomycetia, coloured to indicate the number of genomes positive for Umb toxin particle loci. Within Actinomycetales, only those families containing *umb* loci are listed, with the number of *umb*-containing genomes in parentheses. Asterisks indicate families for which representative *umb* loci are shown in Supplementary Fig. 5. The width of boxes for each family is proportional to the number of species it contains. **b**, Schematic indicating the predicted molecular targets of select toxin domains commonly found in UmbC proteins and representative models for the domains generated using AlphaFold. Models coloured by secondary structure (blue, α-helices; grey, loops and β-strands). Numbers in parentheses indicate the number of UmbC proteins we detected carrying the indicated toxin domain.

Toxin family names are provided in brackets and in Supplementary Table 2. **c**, Predicted structural models of example UmbA proteins selected by virtue of containing multiple distinct or repeated lectin domains (top) or fusions between UmbA and UmbB proteins (bottom). The UmbB domains of bifunctional UmbAB proteins are shown in transparent surface representation and in the same orientation to highlight their conserved interaction with the major cleft of the trypsin-like domain. Concanavalin, UAL-Con-1; Ricin B, Ricin_B_Lectin (Supplementary Table 4). **d**, Model for the intoxication of target cells by Umb toxins, with outstanding questions highlighted. These include the identity of receptor (or receptors) on target cells and the involvement of the lectin domains in mediating binding (1), the role of the stalk in toxin delivery (2) and the mechanism of toxin translocation into target cells (3).

toxins escaped detection for the more than 100 years that scientists have been studying antagonistic interactions among *Streptomyces*[25].

Polymorphic toxins are found in a wide range of organisms, function in many contexts and access their targets through a diverse set of delivery systems[11,16]. Yet, it is difficult to identify a characterized polymorphic toxin system that represents a close analogue of the Umb particle. In certain respects, colicins—antibacterial proteins produced by *Escherichia coli*—might be considered most comparable. Like Umb particles, these are secreted proteins that mediate interactions among closely related strains[26]. However, there are a multitude of features that distinguish colicins and Umb toxin particles, and even their few similarities are superficial. For example, colicins typically target strains that belong to the species of the producer cell, and the diversity of

receptor protein binding domains in colicins (<10) is eclipsed by the diversity of carbohydrate-binding lectin domains associated with Umb particles[27]. Perhaps the starkest of differences between the two polymorphic toxins is their mechanism of secretion, which further highlights their apparently disparate physiological functions. Colicins access the extracellular milieu through a non-canonical mechanism that requires the action of bacteriocin release proteins, referred to as lysis or killing proteins for the death they inflict on producer cells[28]. Colicin expression is thereby under the control of a repressor responsive to cellular damage, and the utilization of these toxins can be categorized as an altruistic behaviour[29]. By contrast, UmbA–UmbC each possess N-terminal Sec (UmbA and UmbB) or TAT (UmbC) secretion signals, and we did not find data to suggest that the release of Umb particles

is detrimental to producer cells. Our work indicates that continued exploration of proteins containing polymorphic toxin domains in diverse bacteria may reveal additional structurally and mechanistically unprecedented toxins.

This work identified the Umb toxin components of *S. coelicolor*, defined their pairwise interactions, revealed the ultrastructure of the particle they form and established the role of these particles in interbacterial antagonism among *Streptomyces* spp. Nevertheless, important open questions for future studies remain. With regard to target cells, it is unclear what roles the UmbA lectin domains have in recognition, the identity of the receptor (or receptors), what role the stalk has and how toxins with cytoplasmic targets cross the membrane (Fig. 5d). In the producer cell, key open questions include how the *umb* genes are regulated, what role immunity proteins have in protection against *cis* and *trans* intoxication, how and where Umb particles assemble, and whether Umb particles from across Actinobacteria are universally used to mediate interbacterial antagonism. It is also of interest to consider the potential biotechnological and therapeutic applications of Umb particles. *Mycobacterium tuberculosis* and *Corynebacterium diphtheriae* are two important human pathogens that, as Actinobacteria, are potential Umb targets and for which resistance to traditional antibiotics is of growing concern[30,31]. In summary, our work identified an antibacterial toxin particle with promise to expand our knowledge of the mechanisms, ecological implications and biotechnological applications of interbacterial antagonism.

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

## Methods

### Bacterial strains and culture conditions

A complete list of strains used in this study can be found in Supplementary Table 6, and all strains generated in this study are available upon request from the corresponding author. *Escherichia coli* strain DH5α was used for plasmid maintenance, strain ET12567 (pUZ8002) for interspecies conjugation and strain Rosetta(DE3) for protein expression. *E. coli* strains were grown in Lysogeny broth (LB) at 37 °C with shaking or on LB medium solidified with 1.5% w/v agar. *S. aureus* strain RN4220 was used for plasmid maintenance and protein expression. *S. aureus* was grown in B2 broth, LB supplemented with 0.2% (w/v) glucose (LBG) or on LBG solidified with 1.5% (w/v) agar. Strain *S. coelicolor* A3(2) was used in Umb characterization studies. Unless otherwise noted, this and other *Streptomyces* spp. used were cultivated in R5 or TSBY liquid medium at 28 °C in baffled flasks with glass beads (3 mm diameter) shaking at 220 r.p.m. or on TSB, ISP2, ISP4 or SFM solidified with 1.5% w/v agar. Growth conditions of diverse bacterial species used in the broad Umb sensitivity screen can be found in Supplementary Table 5. Media were supplemented as needed with antibiotics at the following concentrations: carbenicillin (150 μg ml⁻¹, *E. coli*), apramycin (50 μg ml⁻¹, *E. coli* and *Streptomyces*), kanamycin (50 μg ml⁻¹, *E. coli*), gentamicin (15 μg ml⁻¹, *E. coli*), trimethoprim (50 μg ml⁻¹, *E. coli* and *Streptomyces*), chloramphenicol (25 μg ml⁻¹, *E. coli*; 10 μg ml⁻¹, *S. aureus*) and hygromycin (25 μg ml⁻¹, *E. coli*).

### Plasmid construction

Plasmids used in this study, details of plasmid construction and primers used in this work are provided in Supplementary Table 6. Plasmids generated in this study are available upon request from the corresponding author. Primers and synthetic DNA fragments were obtained from Integrated DNA Technologies. All plasmid constructs were designed using Geneious Prime and generated using Gibson assembly, and all constructs were confirmed by sequencing. For heterologous expression of Umb complex proteins in *E. coli*, the genes were amplified and inserted into NcoI-digested and XhoI-digested pET-22b(+) or NdeI-digested and XhoI-digested pET-28b(+) to generate C-terminal or N-terminal hexahistidine fusions, respectively. VSV-G fusions, point mutations and linkers were introduced to genes amplified from the *S. coelicolor* genome through the cloning primers. umbC1(ring) expression plasmids were constructed by amplifying ALF1–ALF4 (residues A46–A241) and ALF5–ALF8 (residues A532–H798) as two DNA fragments with a linker of two GGGGS repeats introduced in the cloning primers.

Plasmids used for the heterologous expression of UmbC1 and UmbD1 in *E. coli* for mutational profiling were pSCrhaB2 and pPSV39-CV, respectively. To generate these plasmids, the genes were amplified from synthesized DNA fragments codon optimized for expression in *E. coli*. Plasmid pEPSA5 was used for the heterologous expression of various *umbC* toxin domains in *S. aureus*. The toxin domain was either inserted into digested plasmid alongside a N-terminal 3×Flag tag fragment or alongside a signal-sequence-containing 3×Flag tag fragment, with a N-terminal 3×Flag tag being introduced through the cloning primers. These Gibson reactions were transformed into *S. aureus* RN4220 through electroporation, and transformants were maintained in LB supplemented with 0.2% w/v glucose (to repress toxin expression) and chloramphenicol.

*S. coelicolor* genetic manipulation was conducted using a derivative of the suicide vector pKGLP2 (ref. 32), in which the hygromycin-resistance cassette (*hyg*) was replaced with the apramycin resistance gene (*aac*(3)*IV*) and the promoter from pSET152 (ref. 33). This plasmid, pKGLP2a, was generated by amplifying the vector backbone of pKGLP2 and the apramycin resistance cassette from pSET152 by PCR and combining by Gibson assembly. Constructs for introducing deletions, epitope tags and point mutations in the *S. coelicolor* genome with pKGLP2a were generated using Gibson assembly of 1.5–2 kb arms

flanking the site of modification. Complementation of the *umbC2* mutation in *S. coelicolor* was performed using pSET152, into which *umbC2* and its native promoter were cloned using Gibson assembly.

### Structural modelling of Umb proteins and PPIs

Structural predictions for UmbC1–UmbC3, UmbA1–UmbA5 and UmbB1–UmbB3 were made using AlphaFold2 (ref. 18). MSAs were generated by running hhblits[34] against UniRef30 (ref. 35) and BFD[36]. These multiple sequence alignments (MSAs) were uploaded to Colab-Fold[37] and a total of five AlphaFold predictions were generated for each target. Only UmbC3 generated predictions that were consistent with the cryo-EM density of the protein, whereas models for UmbC1 and UmbC2 all resulted in the long coiled-coil folding back on itself. This result prompted the decision to use the UmbC3 model as a template structure for predicting UmbC1 and UmbC2, which enabled the generation of models with a straight coiled-coil consistent with the cryo-EM density. The models with highest predicted local distance difference test (lDDT) were selected for each.

RoseTTAFold2 (ref. 38) was used to predict UmbA–UmbB protein complex structures. MSAs were generated as described above for UmbC1–UmbC3. Paired MSAs for all UmbA–UmbB pairs were generated by matching taxonomy identifiers according to previously published methods[39]. These paired MSAs were provided as inputs to RoseTTAFold2 and produced confident predictions in all cases (predicted lDDTs > 0.8). A similar method was used to compute predictions for interactions between UmbB and individual ALF repeats of UmbC1–UmbC3. In brief, MSAs were generated for UmbB1, UmbB2, UmbB3, UmbC1, UmbC2 and UmbC3 by running HHblits[34] against Uniref30 and BFD, and paired MSAs for all three pairs were generated by maxing taxonomy identifiers. Then, predictions were made for each UmbB model against each of the eight ALF repeats of the corresponding UmbC model. Rather than regenerating the MSA for individual repeats, the paired full-length MSA was trimmed over the region of each repeat.

Owing to the availability of cryo-EM data, models for UmbC1–UmbB1 were generated first. Three different variants of repeat modelling were attempted: (1) trimming to exactly the two-helix repeat; (2) extending by five residues on either side of the repeat; and (3) extending by ten residues on either side of the repeat. To evaluate each modelling variant, the predicted structure and predicted interface error of the UmbC–UmbB interface[18] were considered. All three trimming approaches produced results consistent with the EM data, but the most distinct signal in terms of interfacial predicted interface error was achieved by adding in ten residues of padding. This strategy was applied to UmbC2–UmbB2 and UmbC3–UmbB3.

### Construction of genetically modified *Streptomyces* strains

The pKGLP2a suicide plasmid was used to generate genetically modified *S. coelicolor* strains, including gene deletion mutants and strains expressing chromosomally encoded, epitope-tagged proteins as previously described[32], with modifications described below. Genetic modification constructs were transferred to *S. coelicolor* by intergeneric *E. coli*–*Streptomyces* conjugation using donor strain *E. coli* ET12567 (pUZ8002) as previously described[40]. In brief, overnight cultures of *E. coli* ET12567 (pUZ8002) harbouring the plasmid to be transferred were grown in LB supplemented with chloramphenicol, kanamycin and apramycin. These cultures were washed, concentrated and combined with *Streptomyces* spores following a 10-min 50 °C heat-shock treatment. The mixture was plated on SFM medium supplemented with 10 mM MgCl₂ and incubated at 30 °C for 16–20 h. The plate was then overlaid with 1 ml sterilized dH₂O supplemented with trimethoprim and apramycin. Incubation was continued at 30 °C until transconjugants appeared and were restreaked onto medium supplemented with trimethoprim and apramycin. Confirmed transconjugants were grown in non-selective TSBY medium for about 36 h. These cultures were then restreaked on non-selective SFM agar and incubated at

30 °C for 7 days to produce spores. Spores were then collected, diluted and plated on SFM agar supplemented with 50 mg l$^{-1}$ 5-bromo-4-chloro-3-indolyl-b-D-glucuronide. After incubation for 36 h, white colonies were screened for the presence of the desired allele by PCR. Apramycin-resistant *S. griseus* and complemented *S. coelicolor* Δ*umbC2* were generated through intergeneric transfer of the integrative vector pSET152 or pSET152::*umbC2*, respectively, delivered through conjugation in a similar manner to pKGLP2a.

### IP–MS analysis of UmbC-interacting proteins from *S. coelicolor*

Spores of *S. coelicolor* strains containing *umbC1*–V, *umbC3*–V or *umbA1*–V at the native loci were inoculated in R5 medium and grown for 36 h then back diluted 1:200 in 50 ml R5 medium and further grown for 24–30 h until the optical density at 600 nm (OD$_{600}$) reached 3–4. Spores of *S. coelicolor* containing *umbC2*–V at the native locus were inoculated in 50 ml TSBY medium and grown for approximately 36 h until the OD$_{600}$ reached 4–5. For each strain, 10 ml of the cell culture, including both the cells and culture supernatant, was then mixed with 2.5 ml 5× lysis buffer (750 mM NaCl, 100 mM Tris-HCl pH 7.5, 10% glycerol (v/v), 1 mg ml$^{-1}$ lysosome and 1 mU benzonase). Cells were lysed by sonication and the cellular debris was removed by centrifugation at 35,000*g* for 30 min. VSV-G-tagged proteins were enriched by incubation of cell lysates with 40 µl anti-VSV-G agarose beads at 4 °C for 4–5 h with constant rotation. The agarose beads were then pelleted by centrifugation at 300*g* for 2 min, washed 3 times with 10 ml wash buffer (150 mM NaCl, 2% glycerol (v/v) and 20 mM Tris-HCl pH 7.5) and then washed 3 times with 10 ml 20 mM ammonium bicarbonate. Anti-VSV-G agarose beads and bound proteins were then treated with 10 µl of 10 µg µl$^{-1}$ sequence-grade trypsin (Promega) for 16 h at 37 °C with gentle shaking. After digestion, the agarose beads and peptides were gently mixed and centrifuged at 300*g* for 2 min. After collection of the supernatant, 90 µl of 20 mM ammonium bicarbonate was added to the beads, gently mixed and centrifuged again. The supernatant was collected and combined as the peptide fraction. The mixture was reduced with 5 mM tris(2-carboxyethyl)phosphine hydrochloride for 1 h at 37 °C, followed by alkylation using 14 mM iodoacetamide for 30 min in the dark at room temperature. The alkylation reaction was quenched by adding 5 mM 1,4-dithiothreitol. Acetonitrile (ACN) and trifluoroacetic acid (TFA) were added to the samples for a final concentration of 5% (v/v) and 0.5% (w/v), respectively. Then, the samples were applied to MacroSpin C18 columns (7–70 µg capacity) that had been charged with 100% ACN, LC–MS-grade water and 0.1% TFA. Bound peptides were washed twice with 0.1% TFA and then eluted with 80% ACN with 25 mM formic acid. The dried peptides were dissolved in 5% ACN with 0.1% formic acid and analysed by LC–MS/MS as previously described[41]. Data were analysed using MaxQuant[42], and filtered to remove noise from low abundance proteins with five or fewer spectral counts in IP samples. Enrichment of proteins in the IP samples was determined by dividing the relative abundance of each protein passing the filtering criteria in the IP samples by its relative abundance in the control.

### Purification of heterologously expressed Umb proteins

A subset of the PPI studies and the protease activity assay used purified, heterologously expressed Umb proteins. To purify these proteins, overnight cultures of *E. coli* Rosetta(DE3) carrying pET-22b(+) or pET-28b(+) constructs expressing the protein of interest were back diluted 1:300 in 2×YT broth and grown at 37 °C with shaking at 220 r.p.m. until the OD$_{600}$ reached 0.4. The incubation temperature was reduced to 18 °C, and after 30 min, IPTG was added to a final concentration of 0.3 mM and the cultures were incubated for a total of 18 h. Cells were then collected by centrifugation and resuspended in lysis buffer containing 200 mM NaCl, 50 mM Tris-HCl pH 7.5, 10% glycerol (v/v), 5 mM imidazole, 0.5 mg ml$^{-1}$ lysosome and 1 mU benzonase. Cells were then lysed by sonication and the cellular debris removed by centrifugation at 35,000*g* for 30 min at 4 °C. The 6×His-tagged proteins were purified

from lysates using a 1 ml HisTrap HP column on an AKTA fast protein liquid chromatographer (FPLC). Column-bound protein was eluted using a linear imidazole gradient from 5 to 500 mM. Protein purity was assessed by SDS–PAGE and Coomassie staining. The fractions with high purity were concentrated using 10 kDa cut-off Amicon filters and then further purified by FPLC using a HiLoad 16/600 Superdex 200 pg column (GE Healthcare) equilibrated with sizing buffer (500 mM NaCl, 50 mM Tris-HCl pH 7.5 and 10% glycerol (v/v)).

### PPI assays

Interactions between Umb proteins were probed using proteins heterologously expressed in *E. coli*. For tests of the interactions between UmbB1, UmbA5(T) and UmbC1(ring), 400 µl equilibration buffer (200 mM NaCl, 50 mM Tris-HCl pH 7.5 and 10 mM imidazole) containing 5 µg of purified UmbB1–H, UmbA5(T)–H or UmbC1(ring)–H was mixed with 400 µl *E. coli* cell lysate containing UmbA5(T)–V, UmbC1(ring)–V or UmbB1–V, respectively. To assess input protein levels, 40 µl of these samples was mixed with 4× Laemmli loading buffer (Bio-Rad) and boiled for 20 min at 95 °C for western blot analysis. The remaining protein mixtures were incubated with 50 µl Ni-NTA agarose beads (Qiagen) at 4 °C for 1.5 h with constant rotation. Agarose beads were pelleted by centrifugation at 300*g* for 3 min and washed 5 times with 1.4 ml wash buffer (500 mM NaCl, 50 mM Tris-HCl pH 7.5 and 25 mM imidazole). Proteins bound to the Ni-NTA resin were then eluted with 100 µl elution buffer (500 mM NaCl, 50 mM Tris-HCl and 300 mM imidazole). The eluate was mixed with 4× Laemmli loading buffer, boiled and subjected to western blot analysis. For the competitive binding experiments between UmbB1 and its partners UmbA5(T) and UmbC1(ring), 3 µg of purified UmbB1–H was incubated with 50 µl Ni-NTA agarose beads at 4 °C for 1 h with constant rotation, followed by 2 washes with equilibration buffer. Next, 400 µl equilibration buffer with 2-fold molar excess of purified competitor UmbC1(ring)–H or UmbA5(T)–H was mixed with 400 µl *E. coli* cell lysates containing UmbA5(T)–V or UmbC1(ring)–V, respectively. The protein mixture was further incubated with UmbB1–H bound to Ni-NTA agarose beads and then washed and processed as described above. For the other PPI assays (Figs. 2d–f and 3e and Extended Data Fig. 5a), *E. coli* cell lysates containing 6×His-tagged bait proteins were mixed directly with *E. coli* cell lysates containing VSV-G-tagged target proteins then incubated with Ni-NTA agarose beads, washed and processed as detailed above.

### Western blot analysis

To analyse the PPI assays performed using heterologously expressed Umb proteins, equal volumes of input samples or co-IP samples were resolved using SDS–PAGE then transferred to nitrocellulose membranes (Bio-Rad). Following transfer, membranes were blocked in TBST (10 mM Tris-HCl pH 7.5, 150 mM NaCl and 0.1% w/v Tween-20) with 5% w/v BSA (RPI) at room temperature for 1 h. Primary antibodies (anti-His HRP-conjugated (Qiagen) or anti-VSV-G (Millipore Sigma)) were then added at a dilution of 1:5,000 and incubated at room temperature for 1 h. Blots were then washed four times with TBST, and anti-VSV-G blots were incubated with secondary antibody (anti-rabbit HRP-conjugated (Millipore Sigma) diluted 1:5,000 in TBST) at room temperature for 1 h. Finally, blots were washed four times with TBST and were developed using Clarity Max Western ECL Substrate (Bio-Rad) and visualized using an Invitrogen iBright 1500 imager.

### Trypsin assays

The protease activity of purified UmbA1 and UmbA5 trypsin domains was assessed using universal protease substrate (Millipore Sigma) following the manufacturer's protocol. In brief, 50 µl substrate solution (0.4% casein (w/v)) and 50 µl incubation buffer (0.2 M Tris-HCl pH 7.8 and 0.02 M CaCl$_2$) were combined with 100 µl sample buffer (300 mM NaCl and 50 mM Tris-HCl pH 7.8) containing 500 ng purified protein (UmbA1(T) or UmbA5(T)), 100 ng trypsin (Promega, positive control) or

no protein (blank). The mixture was incubated at 37 °C for 15 min before adding 480 µl stop reagent (5% trichloroacetic acid (w/v)). The samples were further incubated 37 °C for 10 min and centrifuged at 13,000g for 5 min. Next, 400 µl of the reaction mixture was combined with 600 µl assay buffer (0.5 M Tris-HCl, pH 8.8) in a cuvette and absorbance was measured at 574 nm.

## Purification of the Umb1 particle for structural studies

*S. coelicolor* spores expressing UmbA1–8×His from the native locus were inoculated into 30 ml R5 medium and incubated at 30 °C with shaking at 220 r.p.m. for 36 h. Cultures were back diluted 1:200 in 50 ml R5 for a total combined culture volume of 700 ml and incubated for 24–30 h, until the $OD_{600}$ reached 4. Cells were then pelleted by spinning at 21,000g for 45 min and the resulting supernatant was filtered (GenClone 25-229, Vacuum Filter Systems, 1,000 ml PES Membrane, 0.22 µm). Next, 600 ml supernatant was combined with 150 ml 5× lysis buffer (1 M NaCl and 250 mM Tris-HCl pH 7.5) and run over a 1 ml HisTrap FF column on an AKTA FPLC purification system to purify the His-tagged proteins. The bound proteins were eluted using a linear imidazole gradient from 0 to 300 mM. Collected fractions were pooled and concentrated using a 100 kDa cut-off Amicon concentrator until reaching a final volume of 600 µl. The protein sample was further purified by FPLC using a Superose 6 Increase 10/300 GL column (GE Healthcare) equilibrated in sizing buffer (150 mM NaCl, 20 mM Tris-HCl pH 7.5 and 3% glycerol) (v/v)). Each fraction was assessed for purity by SDS–PAGE and silver staining. The fractions with the highest purity and concentration were used for negative-stain EM or cryo-EM.

## Negative-stain EM

Purified Umb1 particles were diluted to 0.01 mg ml⁻¹ and immediately subject to adsorption to glow-discharged carbon-coated copper grids for 60 s followed by 2% uranyl formate (w/v) staining. Micrographs were recorded using Leginon[43] on a 120 KV FEI Tecnai G2 Spirit with a Gatan Ultrascan 4,000 4k × 4k CCD camera at ×67,000 nominal magnification. The defocus ranged from −1.0 to −2.0 µm and the pixel size was 1.6 Å. The parameters of the contrast transfer function (CTF) were estimated using CTFFIND[44]. All particles were picked in a reference-free manner using DoG Picker[45]. The particle stack from the micrographs was pre-processed in Relion[46]. Particles were re-extracted with a binning factor of 4, resulting in a final box size of 80 pixels and a final pixel size of 6.4 Å. The reference-free 2D classification was performed using CryoSPARC[47].

## Cryo-EM sample preparation, data collection and data processing

Cryo-EM grids of the Umb complex were prepared using two separate methods and data were combined during data processing. For the first dataset 3 µl of 0.1 mg ml⁻¹ protein samples was loaded onto freshly glow-discharged lacey grid with a thin layer of evaporated continuous carbon before plunge-freezing using a Vitrobot Mark IV (ThermoFisher Scientific) with a blot force of −1 and 2.5 s blot time at 100% humidity and 22 °C. A total of 18,975 movies were collected at a defocus range between −0.2 and −3 µm. For the second dataset, 3 µl of a 3 mg ml⁻¹ purified Umb1 particle sample was loaded onto freshly glow-discharged R 2/2 UltrAuFoil grids before plunge-freezing using a Vitrobot Mark IV (ThermoFisher Scientific) with a blot force of 0 and 6 s blot time at 100% humidity and 22 °C. A total of 3,942 movies were collected at a defocus range between −0.5 and −2.5 µm.

For both datasets, the data were acquired using a FEI Titan Krios transmission electron microscope operated at 300 kV and equipped with a Gatan K3 direct detector and Gatan Quantum GIF energy filter, operated in zero-loss mode with a slit width of 20 eV. Automated data collection was carried out using Leginon[43] at a nominal magnification of ×105,000 with a pixel size of 0.843 Å. The dose rate was adjusted to 15 counts per pixel per s, and each movie was acquired in super-resolution mode

fractionated in 75 frames of 40 ms. Movie frame alignment, estimation of the microscope CTF parameters, particle picking and extraction were carried out using Warp[48]. Particles were extracted with a box size of 304 pixels with a pixel size of 1.686 Å.

Two rounds of reference-free 2D classification were performed using CryoSPARC[47] to select well-defined particle images. After 2D classification, initial models were generated with ab initio reconstruction in cryoSPARC. The initial models were used as references for 3D heterogenous refinement. Particles belonging to classes with the best resolved umbrella-like morphology were selected. To further improve particle picking, we trained the Topaz picker on Warp-picked particle sets belonging to the selected classes after heterogeneous 3D refinement. The particles picked using Topaz were extracted, and particles were subjected to two rounds of 2D classification and heterogenous 3D refinement in cryoSPARC[47]. Particle picking with Topaz improved the number of unique 2D views. The two different particle sets picked from Warp and Topaz were merged, and duplicate particle picks were removed using a minimum distance cut-off of 60 Å. The particles from both the first and second datasets were subsequently combined. 3D refinements were carried out using non-uniform refinement and the particles were transferred from cryoSPARC to Relion using pyem (https://github.com/asarnow/pyem) to be subjected to the Bayesian polishing procedure implemented in Relion[49]. Subsequent 3D refinements in cryoSPARC used heterogeneous refinements to remove junk particles and non-uniform refinement[50] along with per-particle defocus refinement to produce the final reconstruction at 4.3 Å resolution comprising 386,275 particles. The resulting map showed clear density for the overall quaternary architecture and secondary structure of the Umb1 particle. To further improve the density of each spoke, local refinements were performed using soft masks comprising each ternary complex (UmbC1, UmbB1 and UmbA1) using cryoSPARC[47], which produced final resolutions of 4.0–4.14 Å. The best resolved map that produced the 4.0 Å map after local refinement unambiguously showed that the ALF domain of UmbC, UmbB1 and UmbA1 can be fitted into the density of the local refinement map. Reported resolutions are based on the 0.143 gold-standard Fourier shell correlation (FSC) criterion and FSC curves were corrected for the effects of soft masking by high-resolution noise substitution[51,52]. Local resolution estimation was carried out using cryoSPARC[47].

## Umb1 particle model building and refinement

All models were built and refined by iterating between manual rebuilding and refinement in Coot[53] and Rosetta[54]. For the ALF domain of the UmbC1 with UmbB1 and UmbA1 structure, AlphaFold models were used as a starting point. The relevant segments of the ALF domain with UmbB1 and UmbA1 were built into the locally refined map and the atomic coordinates of the disordered regions were removed. The final model of the ALF domain with the UmbB1 and UmbA1 structure was refined and relaxed with Rosetta, using the 4.0 Å locally refined sharpened and unsharpened maps[54]. For the full Umb1 structure, the AlphaFold model of UmbC1 and locally refined ALF domain with the UmbB1 and UmbA1 structure were used as a starting point to manually rebuild models. The ALF domain, UmbB1 and UmbA1 model from local refinement were fitted into each of the five spoke densities. The final Umb1 complex model including UmbA1 and UmbB1 from each spoke and UmbC1 was subsequently refined and relaxed with Rosetta using sharpened and unsharpened maps[54]. Map figures were generated with dust hidden (size 5) and coloured using the 'color near atom' command (range 10) in ChimeraX.

## UmbC toxicity analysis in *S. aureus*

For analysis of the toxicity of UmbC toxin domains in a heterologous host, toxin domains were cloned into the xylose-inducible plasmid pEPSA5. The deaminase and lipid II phosphatase domains were derived from UmbC1 and UmbC3, respectively, of *S. coelicolor*. The 4TM tox

domain was derived from *Streptomyces anulatus*. Plasmids harbouring the toxin of interest or empty vector were isolated from *S. aureus* and transformed in triplicate into competent RN4220 by electroporation followed by 1 h of recovery in B2 medium at 37 °C 220 r.p.m. Transformations were plated on LBG supplemented with chloramphenicol and 0.2% xylose (w/v) to induce toxin expression. Transformant colonies were enumerated, and transformation efficiencies of empty plasmid and toxin-containing plasmid were computed and compared. The entire experiment was repeated independently with a separate preparation of RN4220 competent cells; data from both replicates are included in Fig. 2a.

### Mutational profiling of *E. coli* expressing the toxin domain of UmbC1

Three *E. coli* strains (MG1655 Δ*ung* pPSV39-CV-*umbD1* pSCrhaB2-*umbC1*, MG1655 Δ*ung* pPSV39-CV-*umbD1* pSCrhaB2 (no insert) and MG1655 Δ*ung* pPSV39-CV-*dddAI* and pSCrhaB2-*dddA* (32641830)) were grown overnight in LB supplemented with 15 μg ml$^{-1}$ gentamycin, 50 μg ml$^{-1}$ trimethoprim and 160 μM IPTG. The cultures were diluted 1:100 into fresh medium without IPTG, incubated until the OD$_{60}$ reached 0.6, then supplemented with 0.2% rhamnose (w/v) for toxin induction. Genomic DNA was isolated from the cultures after 60 min of induction, and sequencing libraries were prepared as previously described[55] and sequenced on an Illumina iSeq. Single-nucleotide variant profiling was performed using previously described analysis methods[55,56].

### Preparation of concentrated supernatant for use in Umb toxicity assays

Spores of *S. coelicolor* wild-type and derivative strains were inoculated in R5 medium and grown for 36 h. The cultures were then back diluted 1:200 in 50 ml R5 medium for a total combined culture volume of 150 ml and incubated for 24–30 h until the OD$_{600}$ reached 4. Cells were then pelleted by centrifugation at 21,000$g$ for 30 min. The resulting supernatant was filtered with a 0.45 μm PES membrane vacuum filter and then concentrated using 100 kDa cut-off Amicon concentrators until reaching a final volume of 3 ml. The concentrated supernatant was run over an Econo-Pac 10DG desalting column (Bio-Rad), aliquoted and stored at −80 °C until use.

### Isolation of bacteria from soil used in Umb toxicity screening

Soil isolate strains used in the broad Umb sensitivity screen were collected from sorghum plants grown at the University of California's Agriculture and Natural Resources Kearney Agriculture Research and Extension Center in Parlier, CA, as previously described[57,58]. Root samples were obtained from mature sorghum plants that had been subjected to a prolonged pre-flowering drought. Immediately after extraction of plants from the soil, roots were removed and placed in 25% glycerol (v/v) for 30 min, then placed on dry ice until they were transferred to −80 °C. To remove soil, roots were placed in phosphate buffer and briefly sonicated. They were subsequently vortexed for 60 s in 99% ethanol, 6 min in 3% NaOCl (w/v) and 30 s in 99% ethanol to sterilize the root surface. Roots were washed twice in sterilized dH$_2$O, and 100 μl of rinse water was plated to check surface sterility. Roots were then cut into 1 cm pieces and placed into 2 ml tubes with 25% glycerol (v/v) and incubated for 30 min at room temperature before storing at −80 °C. One 2 ml tube of roots (approximately 200 mg) was thawed and placed in a sterile ceramic mortar with 1 ml PBS buffer. Root tissue was gently ground to release endophytic bacteria into the solution while minimizing lysis of bacterial cells. The solution was serially diluted, and 100 μl dilutions ($10^{-1}$, $10^{-2}$ and $10^{-3}$) were plated onto various media types: ISP2, M9 minimal medium, skim milk, tap water yeast extract and humic acid. Plates were placed at 30 °C, and growth was monitored daily. When colonies were visible, they were picked and streaked onto a fresh plate of ISP2, followed by subsequent streaks if necessary to eliminate contamination, until only a single morphology

was observed. The 16S ribosomal V3-V4 RNA sequences of the isolates were determined by Sanger sequencing.

### Screening diverse organisms for sensitivity to *S. coelicolor* Umb toxins

Strains used in this assay included both isolates obtained from culture collections and a subset isolated in this study from the root endosphere of field-grown sorghum plants (see above); all strains used in the assay, their sources and their growth conditions are listed in Supplementary Table 5. Strains were grown at 30 °C. Optical densities of initial cultures were measured and used to prepare 1 ml samples at an OD$_{600}$ of 0.01 in the appropriate medium for each strain. Next, 90 μl of each sample was transferred in duplicate to adjacent wells in a 96-well plate. To one of these wells, 10 μl of Umb supernatant from wild-type *S. coelicolor* was added. To the other well, 10 μl of Δ*umb* supernatant from *S. coelicolor* Δ*umb* was added. The plates were then incubated in a BioTek LogPhase 600 Microbiology Reader set to incubate the plates at 30 °C with shaking at 800 r.p.m. taking OD$_{600}$ measurements every 20 min for a total of 48 h. Growth curves were monitored for the beginning of exponential phase. When an organism reached the beginning of its exponential growth phase, the corresponding duplicate cultures were removed from the incubator, combined with 100 μl BacTiter-Glo reagent (Promega BacTiter-Glo Microbial Cell Viability Assay) and incubated at room temperature for 7 min. The luminescent signal was measured in a BioTek Cytation 1 imaging reader. Growth inhibition was assessed by calculating the ratio of signal obtained from cultures incubated with Δ*umb* supernatant by that obtained from Umb supernatant-treated samples. Two biological replicates of the screen were formed, and $Z$ scores were calculated from the average of log$_2$-transformed average ratios from across all strains screened.

### Validation of initial hits from the diverse organism Umb sensitivity assay

Potential target strains *S. griseus* NRRL B-2682 and *S. ambofaciens* SAI 195 along with negative control strain *Streptomyces mobaraensis* NRRL B-3729 were grown on SFM plates for 3 days. Colonies from these plates were excised and used to inoculate 30 ml TSBY and incubated for 20 h (*S. ambofaciens* and *S. griseus*) or 36 h (*S. mobaraensis*) before being prepared for the Umb supernatant sensitivity assay as described above. Assay plates were initially incubated in a log phase for 7 h. Samples were then collected, combined with BacTiter-Glo reagent and luminescence was measured every 2–3 h until the plates reached 20 h of total growth. At 16 h, samples of each culture were serially diluted and plated on ISP2 agar to obtain an independent measure of growth yield.

### Assessing the toxicity of Umb supernatant deriving from *S. coelicolor* mutants

The toxicity of supernatant deriving from individual Umb particle mutants was assessed towards the sensitive species *S. griseus*. For these experiments, the concentrated supernatants from wild-type *S. coelicolor*, mutants unable to synthesize individual Umb particles, *S. coelicolor* Δ*umbC2* (pSET152::*umbC2*) and *S. coelicolor* Δ*umbC2* (pSET152) were prepared as described above. Pre-cultured *S. griseus* (grown for 20 h, as described above) was diluted to OD$_{600}$ of 0.01 in TSBY medium, and 90 μl of this was mixed with 10 μl concentrated supernatant in a 96-well plate. Assay plates were incubated in a log phase for 16 h. The samples were then collected, mixed with BacTiter-Glo reagent (Promega) and luminescence measured. Data represent the r.l.u. normalized by the maximum and minimum levels detected across treatments in an assay.

### Streptomyces co-culture competition assays

For growth competition experiments between *Streptomyces* species, *S. coelicolor* spores were first inoculated into two 50 ml TSBY cultures and grown for about 36 h. Apramycin-resistant *S. griseus* was similarly

inoculated in TSBY and grown for 20 h. When *S. coelicolor* cultures reached an $OD_{600}$ of 3, 10 ml was aliquoted into four replicate baffled flasks. *S. griseus* cells were washed twice with TSBY and then added to the culture flasks at $OD_{600}$ of 0.03, establishing an initial *S. coelicolor* and *S. griseus* ratio of 100:1. Cultures were serially diluted and plated on selective (for *S. griseus*) and non-selective medium (total population) for quantification of colony-forming units at an initial time point and after incubation at 28 °C for 12 h.

## Microscopy

Imaging was performed on a Nikon Eclipse Ti-E wide-field microscope equipped with a sCMOS camera (Hamamatsu). A ×60, 1.4 NA oil-immersion PH3 objective was used for imaging. The microscope was controlled using NIS-Elements (v.3.30.02). The microscope chamber was heated to 28 °C, and *S. griseus* spores were loaded into all four chambers of a bacterial microfluidic plate (B04 from EMD Millipore). Using a CellASIC ONIX (Model EV262) microfluidic perfusion system, a pressure of 2 psi was applied to two columns over two roughly 6-h intervals. One chamber was treated with medium and Umb supernatant for interval one (0–370 min) followed by medium alone for interval two (370–660 min). A second chamber was treated with medium and Δ*umbC2* supernatant followed by medium and Umb supernatant. A third chamber was treated with medium alone followed by medium and Umb supernatant. Finally, a fourth chamber was treated with PBS followed by medium alone.

*Z* stacks were acquired at each of the three positions in each imaging chamber every 10 min. *Z* stacks were merged using Gaussian focus stacking followed by automatic frame alignment in Fiji[59]. Cells that were imaged without occlusion or growth outside the field of view for the duration of 11 h were manually selected and exported in napari[60] using the napari-crop and napari-nd-cropper plugins. Cells were automatically segmented frame-by-frame using Omnipose (bact_phase_omni model)[61]. Spurious labels arising from plate defects, debris or pillars were manually removed in napari following automatic edge-based filtering in Python. Finally, cells were tracked (and any oversegmentation resolved) by manually recolouring *Z* stack labels in napari using the fill tool in 3D mode. All processed space–time labels were then loaded into Python for extracting area over time per cell.

## Bioinformatics analysis

To comprehensively retrieve UmbC protein homologues, the PSI-BLAST program[62] was used for iterative searches against the NCBI non-redundant (nr) protein database until convergence, with a cut-off *e*-value of 0.005. The five upstream and five downstream gene neighbours of UmbC were extracted from the NCBI GenBank files for use in the gene neighbourhood analysis[63]. All protein neighbours were clustered based on their sequence similarities using the BLASTCLUST program, a BLAST score-based single-linkage clustering method (https://ftp.ncbi.nih.gov/blast/documents/blastclust.html). Protein clusters were then annotated based on their domain architectures using the HMMSCAN program[64], searching against the Pfam database[65] and our in-house custom HMM profile database. Signal peptide and transmembrane region prediction was determined using the Phobius program[66]. For systematic identification and classification of C-terminal toxin domains in UmbC proteins and the immunity families represented by UmbD proteins, we utilized the CLANS program[67]. This program uses a network analysis to organize sequences through the application of the Fruchterman and Reingold force-directed layout algorithm[68] based on their sequence similarities derived from all-against-all BLASTP comparisons. A representative sequence of the novel domain family served as a seed in PSIBLAST searches to retrieve homologues. Following removal of highly similar sequences by BLASTCLUST, MSAs were built using KALIGN[69], MUSCLE[70] or PROMALS3D[71]. To identify the conserved residues for each domain family, the Chroma program[72] was used to calculate the conservation pattern of the MSA based on different

categories of amino acid physiochemical properties as previously reported[73]. Structural models for representative sequences of each domain family were predicted using AlphaFold2 (ref. 18) and models with the highest predicted lDDT scores were selected. Determination of domain boundaries for each family was guided by both the structure models and the PAE matrix provided by AlphaFold2. Functional predictions for toxin domains belonging to uncharacterized families were generated using DALI[74] and Foldseek[75] searches with representative structural models from each family to identify structurally related proteins with characterized functions. Function predictions were assigned when structurally similar proteins or protein domains (DALI *Z* score > 3, or Foldseek *E* value < 0.01) with known toxin activities were identified.

## Statistics and reproducibility

Significance of differences in transformation efficiency under heterologous toxin expression, growth yields of *S. griseus* in supernatant toxicity experiments (with supernatant from *S. coelicolor* individual *umb* particle mutants) and competitive indices in competitive growth assays were assessed using analysis of variance and two-sided Dunnett's multiple comparison tests. Significance of differences in protease activity between trypsin and UmbAT proteins and in growth yields from *S. coelicolor* Umb and Δ*umb* supernatant toxicity assays were determined using two-tailed *t*-tests. Tests were performed using Graph-Pad Prism. All western blot assays and were replicated independently a minimum of two times. For bacterial growth assays, the number of replicates collected from independent cultures grown in parallel on a single day are indicated in the figure legends. Each experiment was also replicated at least once on separate days with three additional independent cultures. Statistical methods were not used to predetermine sample size, and blinding and randomization were not employed.

## Reporting summary

Further information on research design is available in the Nature Portfolio Reporting Summary linked to this article.

## Data availability

The cryo-EM maps and atomic structures have been deposited into the Protein Data Bank (PDB) and/or Electron Microscopy Data Bank (EMDB) under accession codes PDB 8W20, PDB 8W22, EMBD EMD-43736 and EMBD EMD-43737. Bacterial protein sequences used for assessing the diversity and distribution of Umb toxins were obtained from the NCBI non-redundant (nr) protein database (https://www.ncbi.nlm.nih.gov/protein/). Source data are provided with this paper.

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

**Acknowledgements** We thank S. Dove, J. Woodward, E. P. Greenberg and C. Harwood for discussions; R. Rodriguez for help with running and troubleshooting MS samples; L. Bai and X. Wang for providing the plasmids pSET152 and pKGLP2, respectively; E. Brodie for providing wheat rhizosphere isolates; and staff at the USDA-ARS Culture Collection (NRRL) for providing strains. This study was supported by Defense Advanced Research Projects Agency Biological Technologies Office Program: Harnessing Enzymatic Activity for Lifesaving Remedies (HEALR) under cooperative agreement no. HR0011-21-2-0012 (to J.D.M.), the National Institute of Allergy and Infectious Diseases (75N93022C00036 to D.V.), a Pew Biomedical Scholars Award (D.V.), an Investigators in the Pathogenesis of Infectious Disease Awards from the Burroughs Wellcome Fund (D.V), the University of Washington Arnold and Mabel Beckman cryo-EM center, National Institutes of Health grant S10OD032290 (to D.V.), a Saint Louis University Startup Fund (to D.Z.), the US Department of Agriculture (CRIS 2030-21430-008-00D to D.C.-D.), and USDA-NIFA (2019-67019-29306 to D.C.-D) and is a contribution of the Pacific Northwest National Laboratory (PNNL) Secure Biosystems Design Science Focus Area 'Persistence Control of Engineered Functions in Complex Soil Microbiomes' (operated by the US DOE under contract DE-AC05-76RL01830 to D.C.-D.). J.D.M. and D.V. are HHMI Investigators, D.V. and J.D.M hold the Hans Neurath Endowed Chair in Biochemistry and the Lynn M. and Michael D. Garvey Endowed Chair in Gastroenterology, respectively, at the University of Washington.

**Author contributions** S.B.P. and J.D.M. conceived the project. Q.Z., S.B., Y.-J.P., K.J.C., D.Z., S.B.P., D.V. and J.D.M. conceived and designed experiments. Q.Z. and S.B. generated bacterial mutant strains and DNA constructs. Q.Z. and Y.W. performed IP–MS experiments. Q.Z. and S.B. performed PPI studies and purified the Umb1 protein complex. Y.-J.P. and D.V. determined the cryo-EM structure of Umb1. F.D. performed structure modelling studies. Q.Z., S.B., K.L.A. and L.A.G. performed experiments defining the physiological function of Umb toxins. C.F.-G. obtained soil isolates for toxicity testing. Q.Z. and K.J.C. performed microscopy experiments. Y.T. and D.Z. performed bioinformatics analyses to define the distribution of Umb proteins. Q.Z., S.B., Y.-J.P., Y.T., K.J.C., P.S., K.L.A., L.A.G., Y.L., F.D., D.Z., S.B.P., D.V. and J.D.M. analysed data. S.B.P. and J.D.M. wrote the manuscript. Q.Z., S.B., Y.-J.P., D.C.-D., F.D., D.Z. and D.V. edited and contributed methods, and all authors provided input. D.C.-D, D.Z., S.B.P., D.V. and J.D.M. supervised work and provided funds and resources.

**Competing interests** The authors declare no competing interests.

**Additional information**
**Correspondence and requests for materials** should be addressed to Joseph D. Mougous.

a

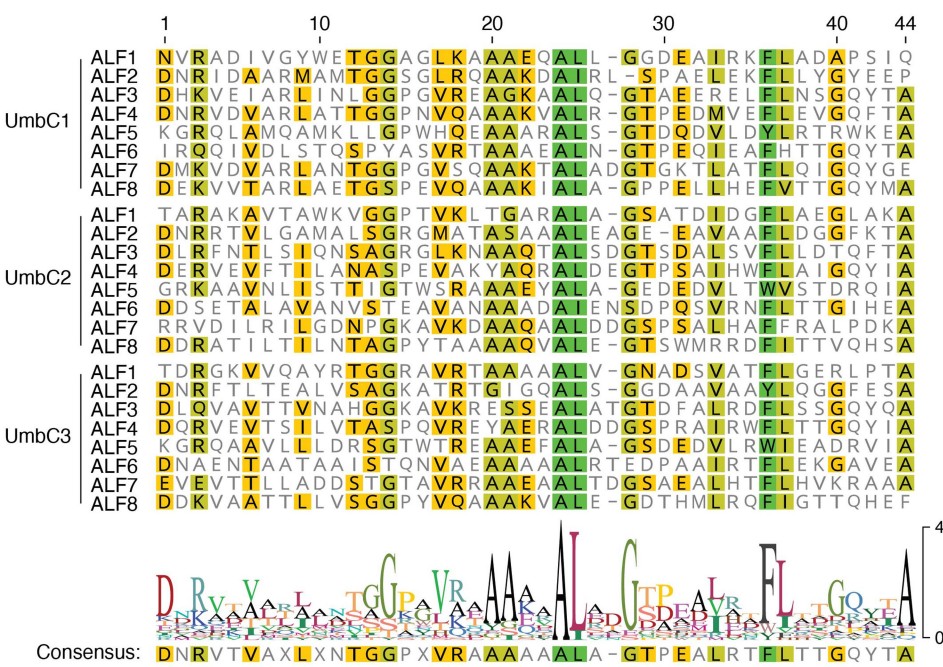

b

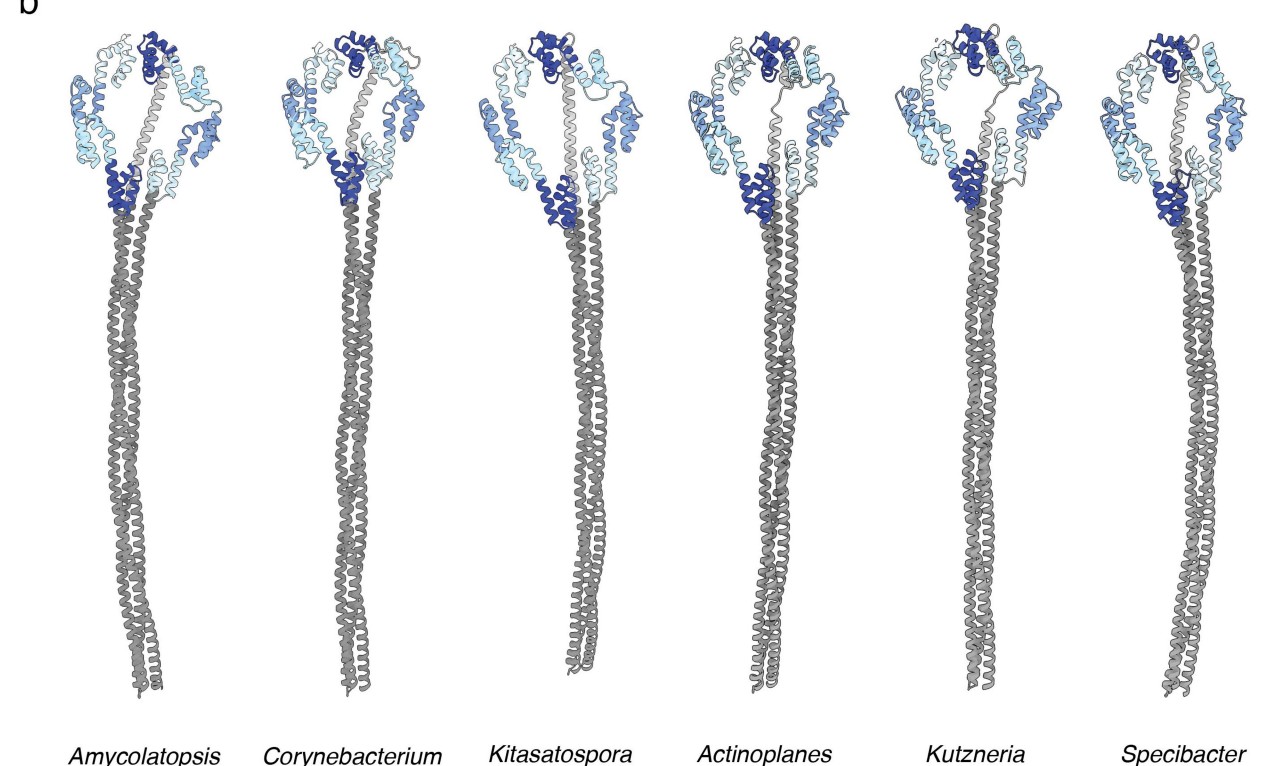

| *Amycolatopsis vastitatis* OXM60336.1 | *Corynebacterium rouxii* VZH84109.1 | *Kitasatospora kifunesis* WP_246561508.1 | *Actinoplanes eccanensis* GID78608.1 | *Kutzneria buriramensis* REH35751.1 | *Specibacter cremis* WP_125616776.1 |

**Extended Data Fig. 1 | Degenerate nature of ALF repeat sequences and example UmbC structural models with straight coiled-coil domains.**
**a**, Alignment of ALF repeats 1-8 from each UmbC protein of *S. coelicolor*. The minimum ALF repeat unit was selected based on the structural model.
**b**, Predicted structural models of assorted UmbC proteins, obtained using default AlphaFold parameters and without templating.

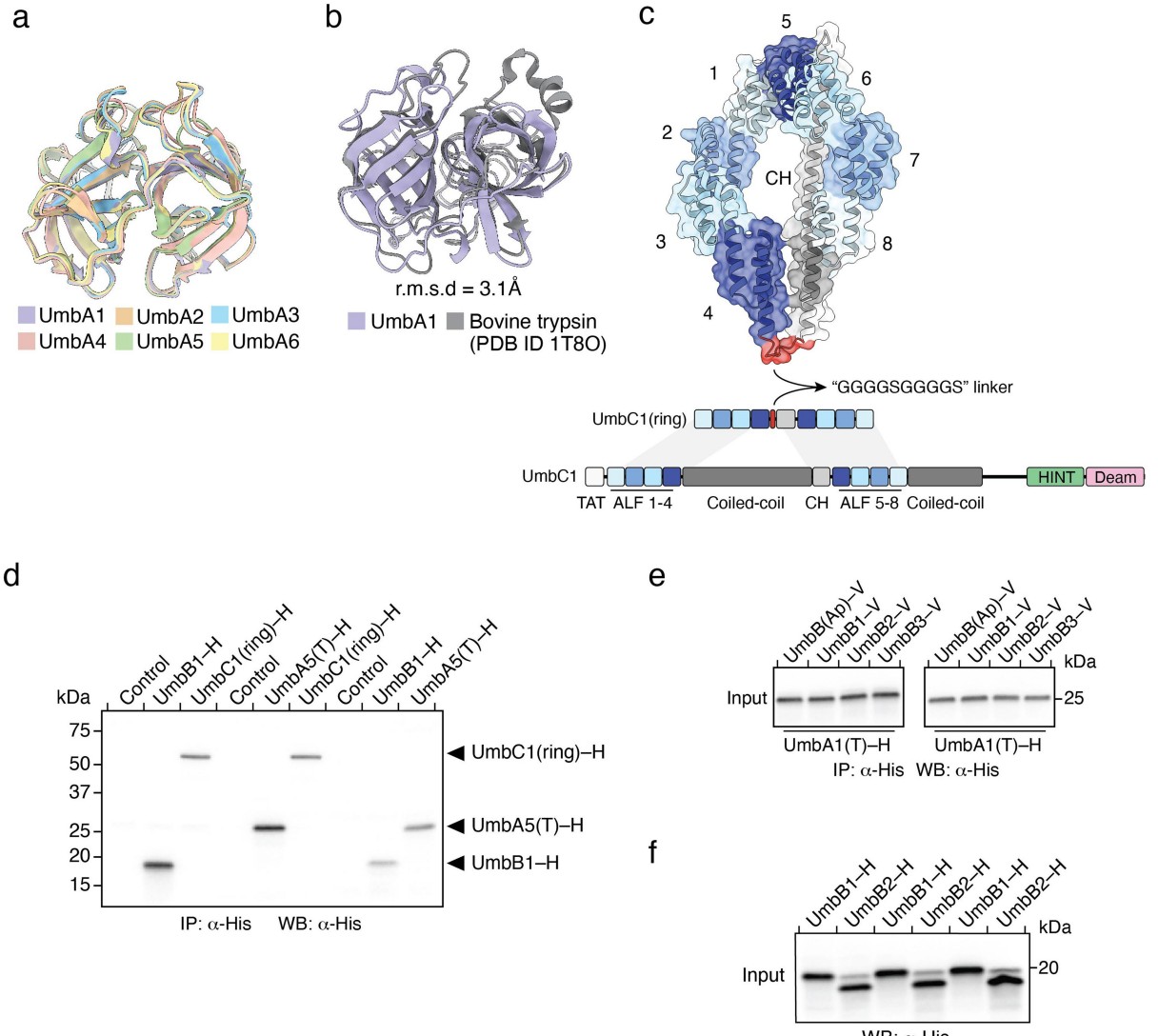

**Extended Data Fig. 2 | UmbA proteins contain a conserved trypsin-like domain, and construct design and input protein levels for studies of the interactions between proteins in the Umb complex. a**, Alignment of the trypsin-like domain of the UmbA proteins of *S. coelicolor.* **b**, Alignment of UmbA1(T) and bovine trypsin. **c**, Predicted structure and genetic architecture of our construct for the expression of UmbC1(ring). **d-f**, WB analyses of input samples from IP experiments between the indicated heterologously expressed, tagged (−H, hexahistidine; −V, VSV-G epitope) Umb proteins. Control lanes correspond to beads in the absence of a bait protein. UmbB(Ap) is a UmbB protein from the distantly related species *Actinoplanes philippinensis.*

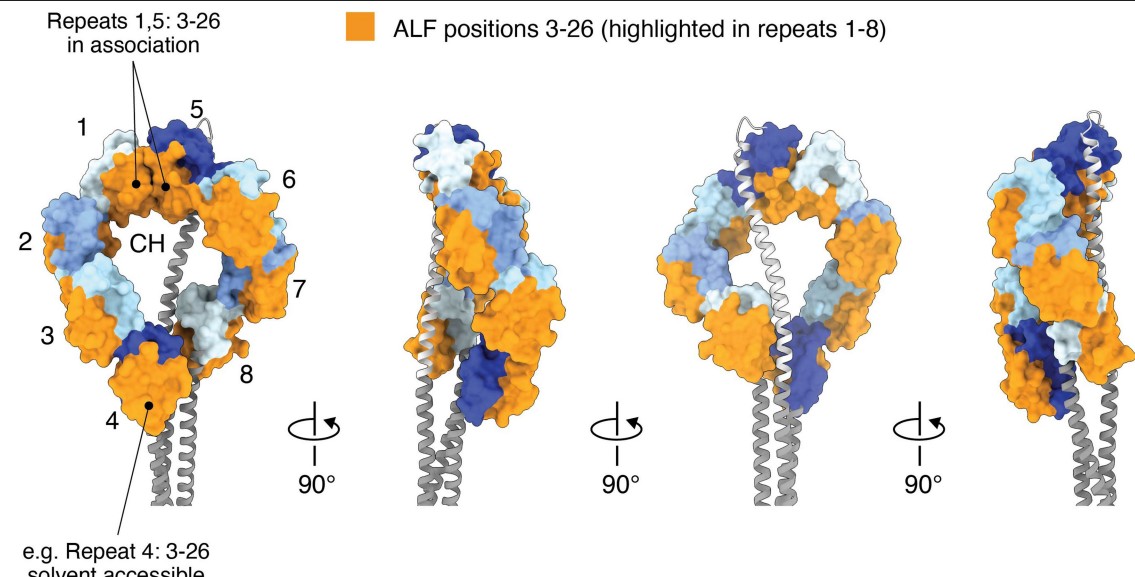

**Extended Data Fig. 3 | ALF repeats 1 and 5 exhibit a distinct orientation.** Orange colouring indicates the residues of the ALF repeats of UmbC1 that are exposed to the surface in repeats predicted to interact with UmbB1 in structural models (ALF 2,3, 4-8). In repeats 1 and 5, many of the equivalent residues are buried in the interface between the ALF repeats.

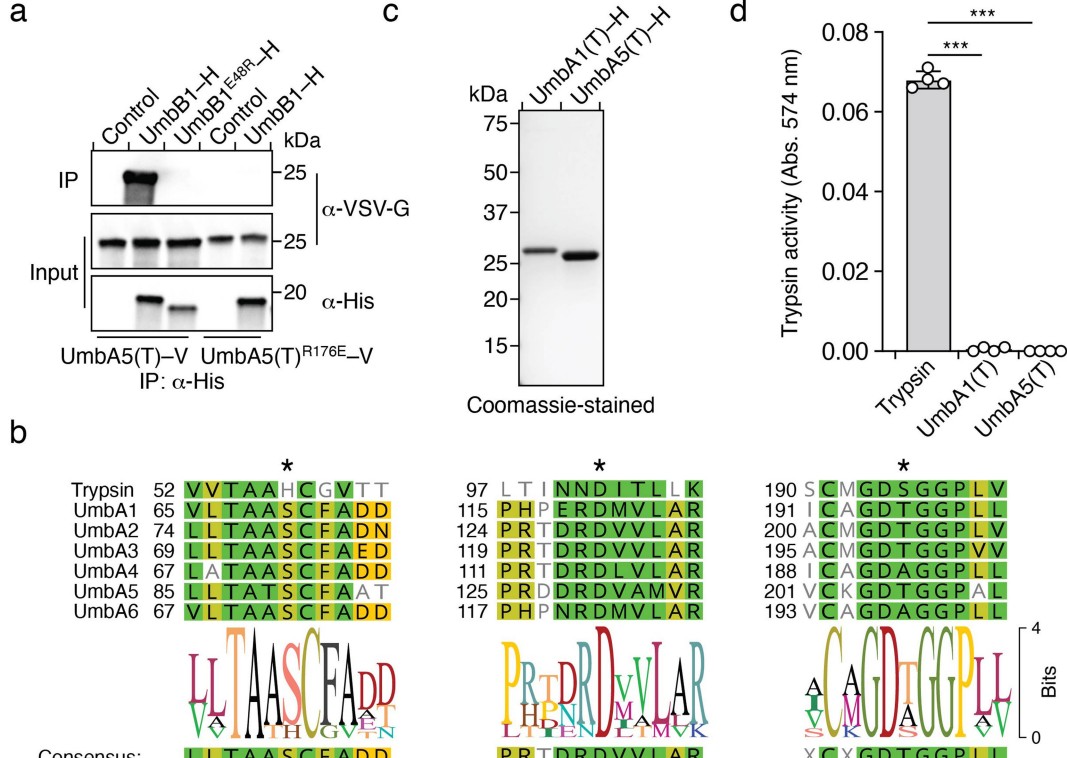

**Extended Data Fig. 4 | The trypsin-like domain of UmbA proteins mediates binding with UmbB and lacks catalytic activity. a**, WB analysis from IP experiments of the indicated heterologously expressed, tagged Umb proteins. UmB1[E48R]–H and UmbA5(T)[R176E]–V contain substitutions of residues predicted to be critical for interaction between the two proteins. Experiment shown is representative of two independent replicates. **b**, Structure-guided alignments of the UmbA(T) regions normally encompassing the catalytic histidine, aspartate, serine triad typical of trypsin proteins, indicating the conserved substitutions found across the UmbA proteins of *S. coelicolor*. **c, d**, Coomassie-stained SDS-PAGE analysis (c) and proteolytic activity (d) of purified, heterologously expressed UmbA1(T) and UmbA5(T). Data in (d) represent mean ± SD (n = 4 technical replicates from one experiment, representative of two biological replicates conducted). Asterisks indicate significant differences from the positive control, porcine trypsin (p < 0.0001, two-tailed t-test).

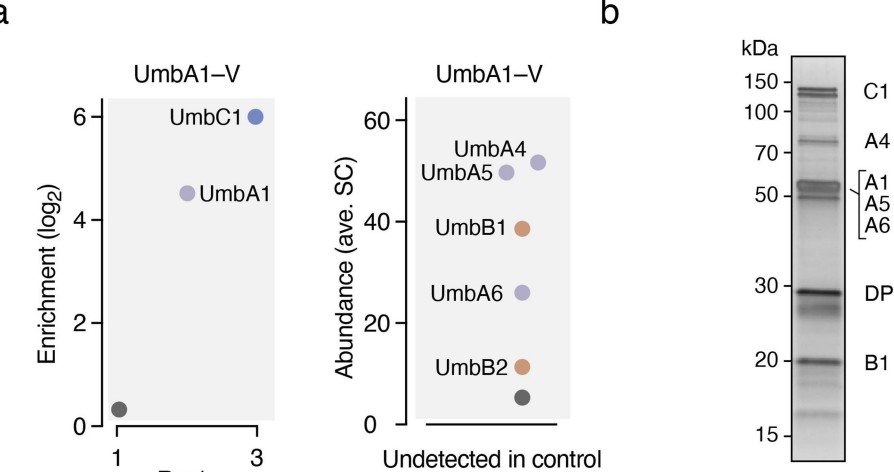

**Extended Data Fig. 5 | Purification of the Umb1 complex using epitope-tagged UmbA1. a**, IP-MS identification of proteins that interact with UmbA1-VSV-G. Left panel indicates the average fold enrichment of proteins detected in both IP and control samples; right panel presents abundance (average spectral counts, SC) for proteins detected only in IP samples. Colours indicate paralogous proteins and correspond to Fig. 2; non-Umb proteins shown in grey, (n = 2 independent experiments). **b**, Silver-stained SDS-PAGE analysis of the Umb1 particle preparation employed in structural analysis, purified using UmbA1–8xHis, with bands corresponding to individual Umb1 proteins identified. DP, degradation product. The experiment is representative of >5 independent replicates.

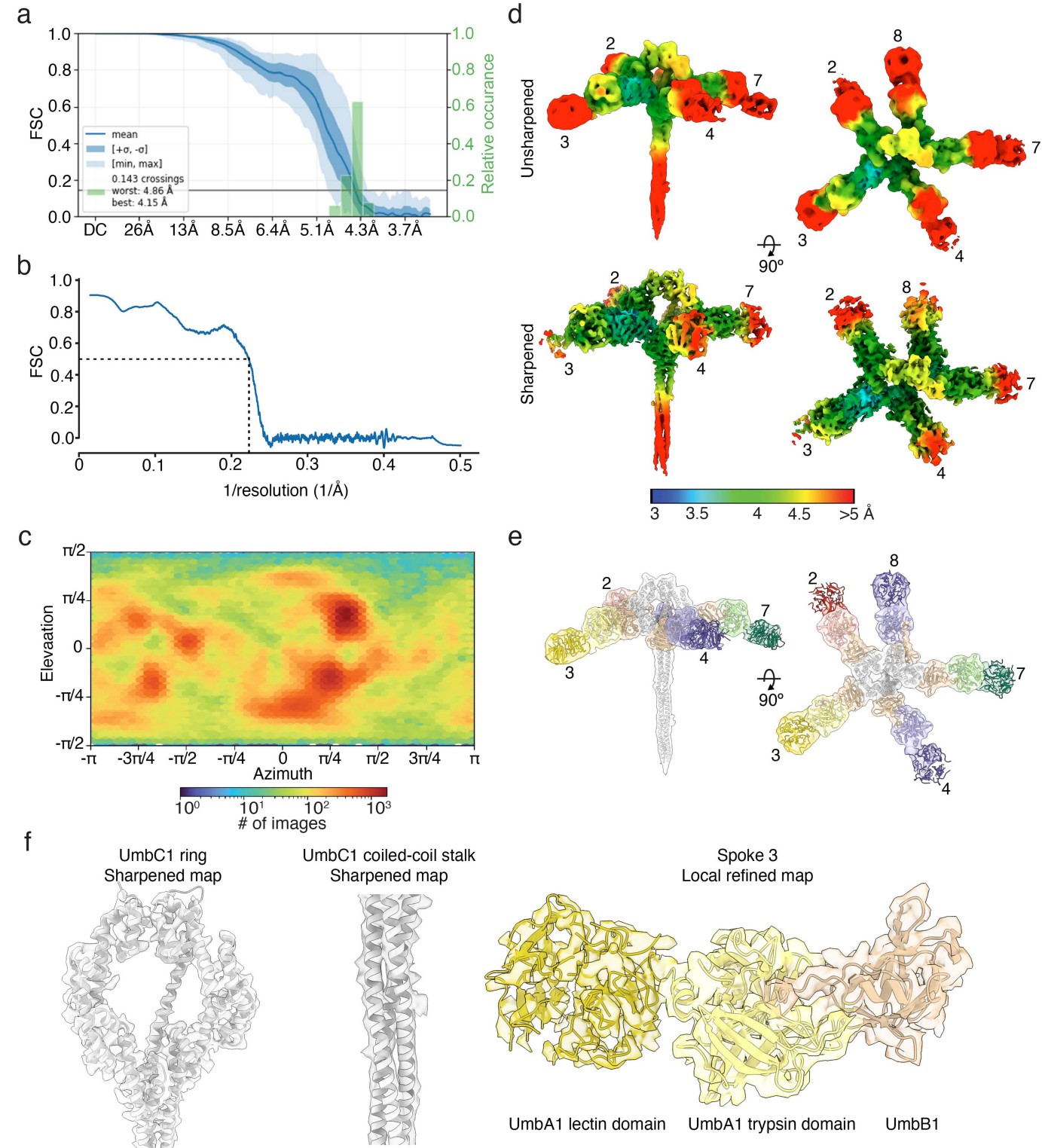

**Extended Data Fig. 6 | Cryo-EM based structural characterization of the Umb1 particle. a**, FSC curve with a cutoff at 0.143 for the overall Umb1 map with the 3DFSC analysis of the full particle. **b**, Map to model FSC curve for the final Umb1 model (in which the lectin domains of UmbA for spokes 2, 4 and 7 were not modeled due to weak density) refined against the Umb1 overall map. **c**, Particle distribution and local resolution of the overall Umb1 map. **d**, Local resolution of the unsharpened (top, threshold of 0.43) and sharpened (bottom, threshold of 1.16) Umb1 cryo-EM map. Numbers correspond to the interacting ALF repeat for each spoke. **e**, The final model within the unsharpened map of the Umb1 particle at a threshold of 0.43. The depiction represents the model derived from our cryo-EM data, with the exception that the lectin domains present in spokes 2, 4, and 7 are shown but were not included in the deposited model due to their weak density. **f**. Close up views of the density corresponding to the UmbC1 ring (threshold of 1.61, sharpened map), UmbC1 stalk (threshold of 1.16, sharpened map), and a single spoke of UmbB1 and UmbA1 (spoke 3, threshold of 1.16, local refined map).

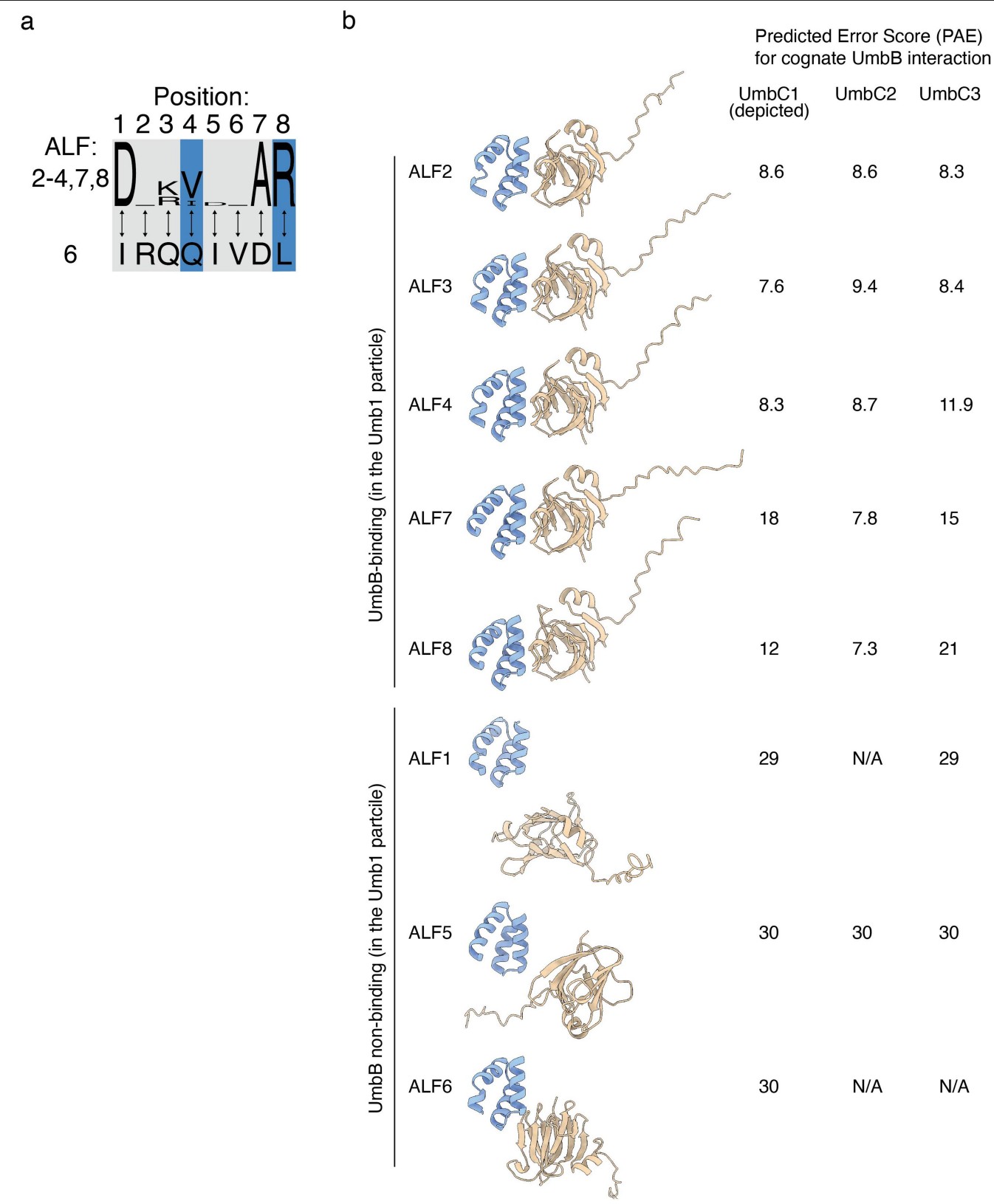

**Extended Data Fig. 7 | Structure and sequence-based differentiation of UmbB-interacting and non-interaction ALF repeats. a**, Probability sequence logo generated from an alignment of positions 1-8 of the UmbB1-interacting ALF repeats of UmbC1, compared to the analogous positions in AFL6. Positions located at the interaction interface and which have non-conservative substitutions in ALF6 are highlighted in blue. **b**, Predicted structural models for the interaction between each ALF repeat of UmbC1 with UmbB1, and RoseTTAFold2 predicted error scores (PAE) calculated for models of the ALF repeats of each *S. coelicolor* UmbC protein interacting with its cognate UmbB. PAE values: <10, high confidence; <20, moderate confidence; >20, low confidence[38]. N/A, no interaction predicted.

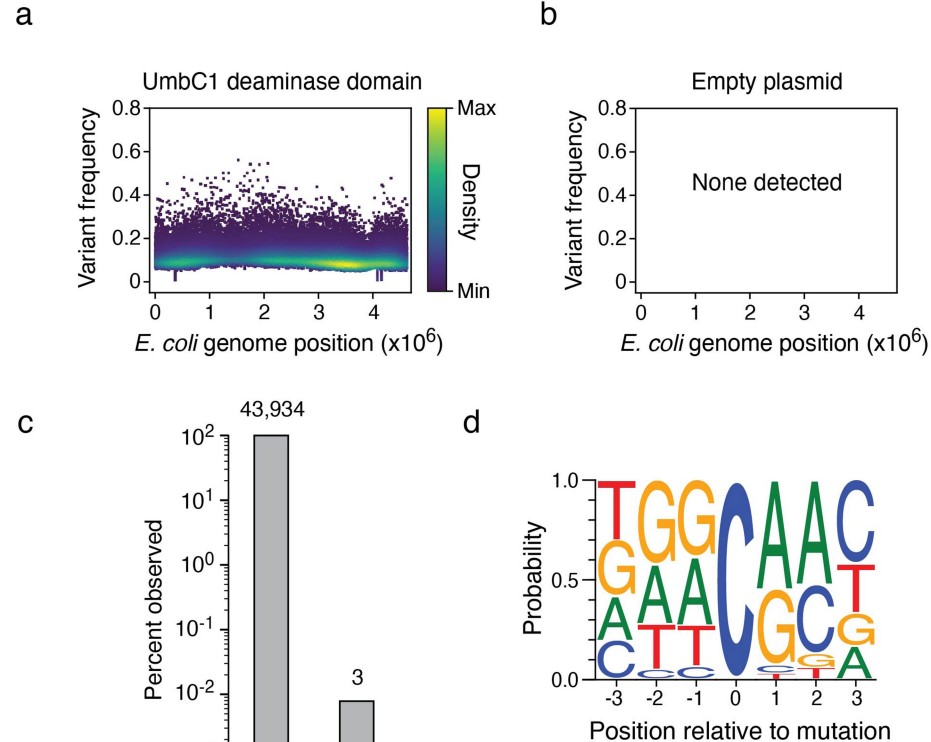

**Extended Data Fig. 8 | The toxin domain of UmbC1 exhibits mutagenic cytosine deaminase activity. a,b**, Representation of single-nucleotide variants (SNVs) by chromosomal position, frequency, and density in *E. coli* Δ*ung* following 60 min induction of expression of the deaminase toxin domain from UmbC1 (a), or the equivalently-treated vector control strain (b). **c**, Frequency of the indicated substitutions among the SNVs shown in (a). **d**, Probability sequence logo of the region flanking mutated cytosines among the SNVs shown in (a).

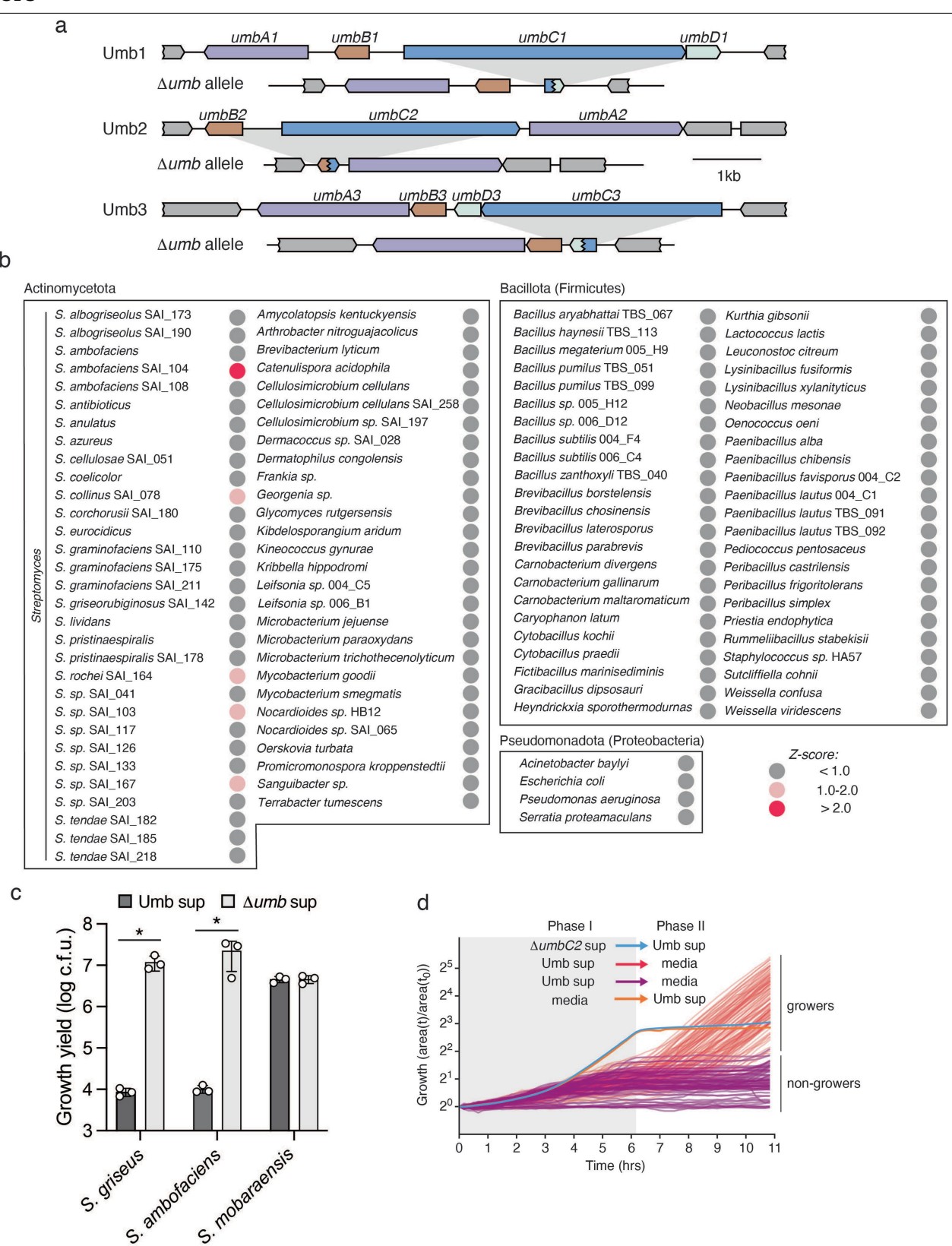

**Extended Data Fig. 9 | Screen of diverse soil bacteria to identify targets of the Umb toxins of *S. coelicolor*. a**, Genetic loci schematic indicating deletions present in *S. coelicolor* Δ*umb*. **b**, Umb toxin target screening results for strains not depicted in Fig. 4b, grouped by target strain phylum. Z-scores were calculated as in Fig. 4b; scores >2 indicate significant Umb-dependent inhibition. **c**, Growth yields (c.f.u, colony forming units) determined of the indicated strains grown in Umb or Δ*umb* supernatant for 16 hr. Data represent means ± SD (n = 3). Asterisks indicate significant differences between the indicated treatments (p < 0.0001 for *S. griseus*, p = 0.0003 for *S. ambofaciens*, two-tailed t-test of log transformed data). **d**, Growth trajectories of individual *S. griseus* cells treated with Umb supernatant. After exchange of Umb supernatant with fresh medium, a portion of the population resumes growth (growers) while other treated cells remain arrested (non-growers). Average growth of Δ*umbC2* supernatant or media-treated populations switched to Umb supernatant treatment in Phase II shown for comparison.

**Extended Data Table 1 | Cryo-EM data collection, refinement and validation statistics**

|  | Full refinement (EMDB-43736) (PDB 8W20) | Local refinement (EMDB-43737) (PDB 8W22) |
|---|---|---|
| **Data collection and processing** | | |
| Magnification | | |
| Voltage (kV) | 300 | |
| Electron exposure (e–/Å$^2$) | 60 | |
| Defocus range (μm) | -0.2 - -3.0 | |
| Pixel size (Å) | 0.843 | |
| Symmetry imposed | C1 | |
| Initial particle images (no.) | 2,716,888 | |
| Final particle images (no.) | 386,275 | 386,275 |
| Map resolution (Å) | 4.3 | 4.0 |
| FSC threshold | 0.143 | 0.143 |
| | | |
| **Refinement** | | |
| Initial model used (PDB code) | Alphafold-generated | Alphafold-generated |
| Model resolution (Å) | 4.5 | 4.2 |
| FSC threshold | 0.5 | 0.5 |
| Map sharpening $B$ factor (Å$^2$) | -245 | -152 |
| Model composition | | |
| Non-hydrogen atoms | 15,412 | 3,750 |
| Protein residues | 2,814 | 629 |
| Ligands | 0 | 0 |
| $B$ factors (Å$^2$) | | |
| Protein | 117 | 79 |
| Ligand | N/A | N/A |
| R.m.s. deviations | | |
| Bond lengths (Å) | 0.007 | 0.007 |
| Bond angles (°) | 1.069 | 0.990 |
| Validation | | |
| MolProbity score | 1.05 | 0.94 |
| Clashscore | 1.56 | 0.76 |
| Poor rotamers (%) | 0.45 | 0 |
| Ramachandran plot | | |
| Favored (%) | 97.16 | 96.83 |
| Allowed (%) | 2.59 | 2.84 |
| Disallowed (%) | 0.26 | 0.33 |

# Reporting Summary

## Statistics

For all statistical analyses, confirm that the following items are present in the figure legend, table legend, main text, or Methods section.

| n/a | Confirmed | |
|---|---|---|
| ☐ | ☒ | The exact sample size (*n*) for each experimental group/condition, given as a discrete number and unit of measurement |
| ☐ | ☒ | A statement on whether measurements were taken from distinct samples or whether the same sample was measured repeatedly |
| ☐ | ☒ | The statistical test(s) used AND whether they are one- or two-sided *Only common tests should be described solely by name; describe more complex techniques in the Methods section.* |
| ☒ | ☐ | A description of all covariates tested |
| ☒ | ☐ | A description of any assumptions or corrections, such as tests of normality and adjustment for multiple comparisons |
| ☐ | ☒ | A full description of the statistical parameters including central tendency (e.g. means) or other basic estimates (e.g. regression coefficient) AND variation (e.g. standard deviation) or associated estimates of uncertainty (e.g. confidence intervals) |
| ☐ | ☒ | For null hypothesis testing, the test statistic (e.g. *F*, *t*, *r*) with confidence intervals, effect sizes, degrees of freedom and *P* value noted *Give P values as exact values whenever suitable.* |
| ☒ | ☐ | For Bayesian analysis, information on the choice of priors and Markov chain Monte Carlo settings |
| ☒ | ☐ | For hierarchical and complex designs, identification of the appropriate level for tests and full reporting of outcomes |
| ☒ | ☐ | Estimates of effect sizes (e.g. Cohen's *d*, Pearson's *r*), indicating how they were calculated |

*Our web collection on statistics for biologists contains articles on many of the points above.*

## Software and code

Policy information about availability of computer code

| Data collection | CryoEM and negative stain EM data collection were carried out using Leginon v3.5. Fluorescence and phase-contrast microscopy data was collected using NIS-Elements v3.30.02. |
|---|---|
| Data analysis | Negative stain EM data processing: CTFFIND v4, DoGPicker v1, Relion v3.0, CryoSPARC v4.4.0 |
| | CryoEM data processing: Warp v1.0.0, CryoSPARC v4.4.0, Relion 3.1 v3.1, pyem v0.5 |
| | Protein structure modeling and visualization: AlphaFold2, RoseTTAFold2 v2020.08.61146, ColabFold, ChimeraX v1.6.1 |
| | Fluorescence and phase-contrast microscopy data analysis: Omnipose 1.0.6, FIJI 2.14.0, Napari 0.4.18, Python 3.10.9 |
| | Statistical analysis of bacterial growth assays: GraphPad Prism v10.1.1 |
| | Bioinformatic analysis: PSI-BLAST 2.13.0+, BLASTCLUST 2.2.26, HMMSCAN (HMMER 3.3.2 package), Phobius v1.01, CLANS, Fruchterman and Reingold force-directed layout algorithm, KALIGN v3.3.2, MUSCLE v3.8.1551, PROMALS3D, Chroma 1.0, DALI v0.99.95, Foldseek, hhblits, Geneious Prime 2023.2.1 |

For manuscripts utilizing custom algorithms or software that are central to the research but not yet described in published literature, software must be made available to editors and reviewers. We strongly encourage code deposition in a community repository (e.g. GitHub). See the Nature Portfolio guidelines for submitting code & software for further information.

## Data

Policy information about availability of data

All manuscripts must include a data availability statement. This statement should provide the following information, where applicable:

- Accession codes, unique identifiers, or web links for publicly available datasets
- A description of any restrictions on data availability
- For clinical datasets or third party data, please ensure that the statement adheres to our policy

All source data for this study will be provided with this paper or are deposited in a public repository. The cryo-EM maps and atomic structures are under deposition in the Protein Data Bank (PDB) and/or Electron Microscopy Data Bank (EMDB) under accession codes PDB ID 8W20, PDB ID 8W22, EMD-43736 and EMD-43737. Bacterial protein sequences used for assessing the diversity and distribution of Umb toxins were obtained from the NCBI non-redundant (nr) protein database.

## Research involving human participants, their data, or biological material

Policy information about studies with human participants or human data. See also policy information about sex, gender (identity/presentation), and sexual orientation and race, ethnicity and racism.

| | |
|---|---|
| Reporting on sex and gender | Not applicable. |
| Reporting on race, ethnicity, or other socially relevant groupings | Not applicable. |
| Population characteristics | Not applicable. |
| Recruitment | Not applicable. |
| Ethics oversight | Not applicable. |

Note that full information on the approval of the study protocol must also be provided in the manuscript.

# Field-specific reporting

Please select the one below that is the best fit for your research. If you are not sure, read the appropriate sections before making your selection.

☒ Life sciences  ☐ Behavioural & social sciences  ☐ Ecological, evolutionary & environmental sciences

For a reference copy of the document with all sections, see nature.com/documents/nr-reporting-summary-flat.pdf

# Life sciences study design

All studies must disclose on these points even when the disclosure is negative.

| | |
|---|---|
| Sample size | Sample size was not statistically predetermined. For experiments in which statistical analysis was applied, a minimum of three replicates were performed to ensure validity of the analyses. For Western blot analyses, a minimum of two biological replicates were performed as is standard in the field, see for example https://www.nature.com/articles/s41586-023-06506-6#MOESM2. For the cryo-EM experiment, data were collected until the resolution could no longer be improved through further data acquisition. |
| Data exclusions | Proteins with low spectral counts (<6) in IP samples were excluded from the enrichment analysis. |
| Replication | Experiments were independently replicated a minimum of two times, as described in figure legends. All attempts at replication were successful. |
| Randomization | This was not relevant to our study, as experiments were performed using clonal bacterial populations or were in vitro assays of purified proteins or lysates of clonal populations of bacteria. |
| Blinding | Blinding was not relevant to our study, as subjective analysis of the data was not required. |

# Reporting for specific materials, systems and methods

We require information from authors about some types of materials, experimental systems and methods used in many studies. Here, indicate whether each material, system or method listed is relevant to your study. If you are not sure if a list item applies to your research, read the appropriate section before selecting a response.

## Materials & experimental systems

| n/a | Involved in the study |
|---|---|
| ☐ | ☒ Antibodies |
| ☒ | ☐ Eukaryotic cell lines |
| ☒ | ☐ Palaeontology and archaeology |
| ☒ | ☐ Animals and other organisms |
| ☒ | ☐ Clinical data |
| ☒ | ☐ Dual use research of concern |
| ☒ | ☐ Plants |

## Methods

| n/a | Involved in the study |
|---|---|
| ☒ | ☐ ChIP-seq |
| ☒ | ☐ Flow cytometry |
| ☒ | ☐ MRI-based neuroimaging |

## Antibodies

| | |
|---|---|
| Antibodies used | anti-VSV-G Sigma, V4888; anti-His HRP conjugated, Qiagen 34460; anti-rabbit HRP conjugated, Sigma A6154 |
| Validation | All antibodies are commercially available. Validation information is provided in datasheets supplied by the manufacturers. See https://www.sigmaaldrich.com/deepweb/assets/sigmaaldrich/product/documents/343/088/v4888dat.pdf for anti-VSV-G and https://www.qiagen.com/us/knowledge-and-support/product-and-technical-support/quality-and-safety-data/sds-search?l=US&q=800000000487 for anti-His HRP conjugated. |

## Plants

| | |
|---|---|
| Seed stocks | Not applicable. |
| Novel plant genotypes | Not applicable. |
| Authentication | Not applicable. |

