## [Peer Review File · Nature]

Manuscript Title: Streptomyces umbrella toxin particles block hyphal growth of competing species

Reviewer Comments & Author Rebuttals

Reviewer Reports on the Initial Version:

Referees' comments:

Referee #1 (Remarks to the Author):

Zhao et al. describe the discovery and structure determination of antibacterial protein complexes from Streptomyces species. The authors name these complexes umbrella particles. I am not an expert in the biological aspects of this work, but the manuscript appears to describe an important and fascinating finding. Bioinformatic analysis suggests that these umbrella particles are widespread, increasing the importance and impact of the work even further.

The 3D model for the umbrella particle was obtained by combining cryo-EM with alphafold prediction of structures and experimentally determined protein-protein interactions.

Cryo-EM of the antibacterial complexes shows a peculiar, elongated structure with a long stalk and five spokes.

The model is probably correct and the 3D map file (at least the sharpened map) presented looks like a reasonable low-resolution map. However, the way the cryo-EM data is presented makes the map look better than it is. There is enough missing information to raise suspicion in a skeptical reader, which is not in the authors' best interests.

I believe Fig. 3c shows a map calculated from the model, not the experimental map. That needs to be made clear.

The map shown in Fig. 3d appears to have been "cleaned up" – more than just using the "hide-dust" feature of Chimera. With the experimental map provided it is not possible for me to find a threshold where the stalk helices and the domains at the top of the structure are shown clearly at the same time. Has the map in Fig. 3d been shown at multiple thresholds? Is it a composite map? The details of what is shown need to be clear.

The methods section for the manuscript describes 2D classification as part of the image analysis process but 2D classes from the cryo-EM images are not shown anywhere (2D classes in Fig. S6C are from negative stain). Please show these 2D class averages from cryo-EM.

The only FSC curves shown are for map-vs-model. Why is a standard halfmap-vs-halfmap FSC curve not

shown? This curve is necessary to believe that the map is valid.

It is hard to imagine that a particle like this one would show isotropic orientations on the cryo-EM grid. Please provide an orientation distribution plot, 3D FSC plot. It is also hard to imagine that resolution in the map is uninform. Please provide a map of local resolution.

I think the structure is probably valid but the details of the structural analysis should be shown with more transparency.

Referee #2 (Remarks to the Author):

Congratulations to Dr. Mougous and colleagues on a fantastic set of discoveries. This manuscript reports the discovery of an entirely novel family of large protein toxins in Actinobacteria. An depth look at examples of these toxins from *Streptomyces coelicolor* shows that the toxins rapidly arrest the vegetative mycelial growth of competitors, revealing a fascinating new mechanism of competition within the complex environment of the soil. Further, the toxins are assembled with a remarkable set of structural building blocks, revealing architectures heretofore unobserved in nature. This paper is paradigm shifting and opens up a new field with a wealth of new possibilities for understanding microbial soil ecology, novel antibiotic strategies, and principles of evolution.

The investigators have performed a tour-de-force study with an integrated and rigorous application of genomics, alpha-fold, cryo-EM, protein biochemistry, and microbiology . The data quality and data presentation are outstanding and allow for a robust set of conclusions. While the need for statistical analyses was limited, it was applied appropriately when needed.

I do not have any substantive suggestions for further improving this manuscript. It was masterfully written with an amazing attention to detail and clarity.

Minor points:

Line 338, typo in spelling of altruistic.

Line 351, in the list of open questions, the authors could also mention the presence of immunity proteins in some strains as this was mentioned early but not readdressed.

Line 355, typo in spelling of diphtheria

Referee #3 (Remarks to the Author):

This paper reveals the existence of a novel class of secreted antibacterial protein complexes made by

Streptomyces, termed Umbrella (Umb) Toxin Particles. The particles consist of a long coiled-coil capped by a ring of toxin repeats, bearing 5 spokes with lectin tips. Bioinformatic analysis shows that these toxin particles are encoded not just within the Streptomycetaceae but also in 6 other actinobacterial orders. The potential diversity of these particles is striking. They incorporate lectin domains via promiscuous UmbA binding, and these lectins are highly variable in nature. In addition, at least 77 types of predicted toxin domains are associated with the UmbC proteins identified bioinformatically, and these toxins have very diverse predicted cellular targets (pore-formers, rRNases, mRNases, DNA deaminases, etc, etc). The authors screen a panel of 140 bacteria, biased toward Streptomyces spp., to look for species sensitive to Umb toxins. Through this route, they identify 2 target bacteria. They show that Umb particle-enriched supernatant from *S. coelicolor* selectively inhibited the growth of 2 Streptomyces species, *S. griseus* and *S. ambofaciens*, but not the growth of the other Streptomyces species included in the screen. In the case of *S. griseus*, time-lapse imaging showed that particle-enriched supernatant caused immediate growth arrest in actively growing vegetative mycelium but did not affect germination. Genetic inactivation of *S. coelicolor* umbC2, but not umbC1 or umbC3, produced particle-enriched supernatant that lacked the ability to inhibit *S. griseus*, consistent with Umb2 particles being responsible for the toxicity. The authors propose that the variable UmbA lectin domains mediate selective target cell binding, but do not address this question experimentally.

The Umbrella Toxin Particles described here represent an important discovery. Unsurprisingly, this work raises many additional biological and mechanistic questions, most of them enumerated by the authors themselves in the closing paragraph of the Discussion. How do the toxins with cytoplasmic targets cross the cell envelope, do the UmbA lectin domains determine target cell recognition, and what are the receptors on the target cells? Nevertheless, given this is the first description of these novel toxin particles, I think the content of the current paper represents a sufficient advance to be considered further for publication. However, there are key issues concerning the structural biology, for which there is a significant lack of detail that prevents proper scrutiny. There is also a troubling blurring of AlphaFold prediction into cryo-EM 'structure'. It is important that these issues are resolved.

BIOLOGY

1. The competition assays were done in liquid. Is there a similar inhibitory effect on vegetative growth on solid medium?
2. Why was the screen of 140 potential target bacteria done with an undefined particle-enriched supernatant, rather than with multiple purified and defined Umb particle preparations?
3. If the culture supernatant from *S. coelicolor* is treated with a protease, does this abolish the inhibitory effect on the growth of *S. griseus*?
4. In the Introduction, in the context of known protein toxin systems in Streptomyces, it might be appropriate to reference the recently described cytoplasmic contractile injection systems (CIS) and their associated protein toxins, which appear to be involved in developmentally regulated cell death. Casu et

al. (2023) Nat Microbiol. 8:711-726. Vladimirov et al. (2023) Nat Commun. 14:1469.

STRUCTURAL WORK

5. Line 180: 'Relative abundance in mass spectrometry data' is insufficient evidence for assignment of subunits in a cryo-EM reconstruction. Were there features in the cryo-EM map that supported the assignment? If so, these need to be presented in the Extended data so that the level of confidence can be assessed by readers. Likewise, were the assigned positions of UmbA4, A5 and A6 clearly identifiable from the density, or assigned arbitrarily? If the latter, there is a significant risk of implying the confidence in the presented model is greater than it is.

6. Line 170: The reason for the limited resolution requires clarification. 'Protein aggregation' presumably implies few particle images were usable as the majority were overlapping? 'Heterogeneity' could imply conformational or compositional variation between particle images. Rather than only being a limit to attainable resolution, this could hold important information on natural variation in the sample composition and/or possible dynamics associated with molecular mechanism.

7. There is a significant lack of detail regarding the cryo-EM data processing workflow and the following omissions need to be dealt with to allow reviewers to make a full assessment of the work. The number of particles (initial and final) is not presented in either the Methods or Extended Data figure legend. The classification of particle images leading the final reconstruction should be presented in the typical format of a 'tree' in the Extended Data where class features are indicated and selected subsets identified. Other pieces of information are also not provided that are standard practice are (i) reconstruction colored by local resolution, (ii) particle orientation distribution, (iii) representative 2D class averages, (iv) three-dimensional FSC to assess reconstruction quality (e.g. <https://3dfsc.salk.edu/upload/>). This information is crucial for Reviewers to assess whether an improvement on the modest resolution reported is achievable.

8. The authors should consider a plot of reconstruction resolution for a series of given number of particles processed. This would indicate the number of additional particle images expected to be needed to reach a resolution sufficient for confident structural model construction (generally considered to be better than ~4 Angstrom resolution). Can the authors provide an estimate of how much additional data might be needed to achieve this with the current grid preparation conditions?

9. The authors should also consider focused refinement, 3D variability analysis or similar tools for analysis of heterogeneous data. It is likely that a subset of the reconstruction could be improved beyond the reported resolution with such targeted data processing strategies. Can the authors confirm these were attempted and give the results?

MINOR COMMENTS

Line 168: 'structure' should be 'reconstruction' or 'map', as these are the experimental data with a

resolution metric.

Line 271. 'Multiple species' feels like an exaggeration for *S. griseus* + *S. ambofaciens*.

Line 280-281. Is Fig 5b the correct figure to cite here?

Line 352-353. "...mounted universally..." meaning not clear.

TYPOS.

Throughout. Either "*Streptomyces*" (Italics, capital 'S') or "streptomycetes" (no italics, lower case 'S')

Line 355. Typo. *Corynebacterium dipHtheriaE*

Referee #4 (Remarks to the Author):

The manuscript by Zhao and co-authors describes an original, exciting and broadly-significant study identifying and characterising a new class of polymorphic toxins which occur widely in Actinobacteria and which are included within large multiprotein 'umbrella' particles unrelated to any previously-described type of anti-bacterial toxin delivery system. The authors show that these toxins block mycelial growth of competitor species and incorporate diverse anti-bacterial toxin domains, many previously unstudied. Additionally, the so-called umbrella toxins are able to direct their own entry into susceptible target bacteria, perhaps using a modular lectin-like recognition of specific surface carbohydrates, opening up the possibility of future biotechnological or therapeutic utilisation. As would be expected following the discovery of a new type of toxin delivery system, many intriguing questions remain about how the particles are assembled, interact with target cells and deliver their toxin domain to the interior of the targeted cell. These are noted by the authors and will no doubt inspire future research spanning many areas to understand the mechanism and ecological significance of this system.

The research has been conducted and presented to a high standard and an extensive body of data is included. Nevertheless, the authors should address one major concern and a number of minor comments, issues and suggestions as detailed below.

Major issue:

There is one major issue that the authors should address. The authors say that 'based on their relative abundance in our mass spectrometry data' they modelled UmbA1 into two of the five spokes in the cryoEM structure of the particle and UmbA4, UmbA5 and UmbA6 each into one of the spokes. This is not a convincing rationale. Spectral counting mass spectrometry cannot be used to determine the amounts of different proteins relative to each other in a quantitative manner. Even if the spectral

counting data were to be considered in a semi-quantitative way, they still do not support the conclusion. In the UmbC1 IP, A1 gives 53 counts (with 6 in the control), A4 gives 49, A5 gives 40 and A6 gives 17, so not most consistent with 2:1:1:1 as proposed. In the UmbA1 IP, A1 gives 145 counts, A4 gives 51.5, A5 gives 49.5 and A6 gives 26. However, here A1 is the bait protein which would be expected to give a higher and non-stoichiometric signal since non-complexed protein in the cell will also be IP'd; whilst the other three, similar to the previous experiment, give an apparent stoichiometry of 2:2:1, not 1:1:1. The authors should provide further evidence to support their conclusion that the particles contain UmbA1, A4, A5 and A6 at a ratio of 2:1:1:1, and that two of the five spokes of the Umb particle contain A1 whilst the other three contain A4, A5 and A6 as assigned in the structure. Can the authors be certain that individual Umb particles do actually contain all four UmbA proteins simultaneously?

Other minor comments / issues to address:

(1) Extended Data Figure 6. In part a, left hand panel, the point corresponding to UmbC1 appears to be missing (or incorrectly placed further down the panel); according to the spectral counting data in Supplementary Table 3, UmbC1 should be present at around $\log_2 6$ (5.99), whereas no protein was detected with enrichment $\log_2 \sim 1.5$ as is currently plotted for the middle point.

(2) Fig. 3a and Extended Data Fig. 6b. It is not totally clear what the difference is between these two panels/results – does Fig. 3a show the result of the UmbA1-V IP rather than the UmbA1-8xHis IP? This should be clarified in the legend to Fig. 3a. Were the UmbA1-V IPs also done on culture supernatant as for the UmbA1-8xHis IPs? This should be noted (UmbA1-V IPs are not specifically described in the Methods currently).

(3) UmbD does not appear to be isolated from the UmbA1-8xHis and UmbA1-V IPs (unlike the initial UmbC1-V IPs). Does this reflect the fact that the presumed immunity protein is not secreted with the Umb particle, and does this have any implications for the mechanism of self-resistance of the producing cell prior to secretion?

(4) Starting on Page 10, the authors refer to screening for 'Umb targets' and 'toxin targets' but only later explicitly state that by 'target' they mean 'susceptible organism' or 'target organism'. Given that the 'target' of a toxin is typically used and interpreted to mean the direct molecular target or substrate of a toxin, suggest to clarify their use of the word target as 'susceptible target organism' or equivalent on first usage in the text and the legend to Figure 4.

(5) Line 208/209 and legend to Figure 4a. The authors state that heterologous expression of the C-terminal domains of the UmbC proteins leads to a drop in 'bacterial viability', whilst the figure depicts transformation efficiency of the plasmid rather than e.g. viable count. This is easily resolved and clarified if the authors note in the text or the legend that the transformation was done under conditions where expression of the toxin was induced and so number of viable transformants represents a read-out of toxicity.

- (6) Figure 4e. The y-axis title says 'log Sc/Sg' but the axis labels are not the log transformed numbers.
- (7) Figure 5a. Could be useful to provide an indication/key for how the size of the boxes at the branch tips relates to the number of genomes included in that tip.
- (8) Figure 5b. The legend could indicate more clearly that these are predicted functions of the toxin domains, particularly in the case of 4TM_tox where the label 'pore-forming' is not yet supported by experimental characterisation of these toxins or any homologous proteins.
- (9) Line 291. Should be Fig. 5b
- (10) Line 300. Should be Fig. 5c,d
- (11) Methods section starting line 670. Multiple instances of 'u' instead of Greek Mu for 'micro'
- (12) The authors mention a custom script (Line 752) which could be made available through GitHub.
- (13) For the microscopy experiment (Fig 4f, g and Extended Data Fig. 11), it is not clear if the experiment was repeated independently and on how many occasions.
- (14) The authors could comment on why there is not always a gene encoding an immunity (UmbD) protein associated with genes encoding UmbC toxins.

Reviewer comments in BLACK

Author responses in BLUE

We thank the reviewers for their generally enthusiastic response to our manuscript, and for their constructive feedback. In addition to addressing to all points raised by the reviewers as detailed in the reviewer-specific point-by-point responses below, we wish to draw the reviewer's attention to several material improvements we made to the manuscript that were *not* specifically requested. Importantly, each of these additions lend further support to or merely clarify our original conclusions. Briefly, these improvements include:

- 1) Using recently developed AI-based methods for particle identification (Topaz, Bepler *et al.* 2019), we roughly tripled the number of Umb1 particles incorporated into our dataset (Extended Data Table 1). As a result, we significantly improved the cryo-EM map quality and resolution (from 5.1 Å to 4.3 Å), and were able to obtain a 4.0Å resolution of a spoke using local refinement, thereby supporting model building and providing a blueprint of the interactions between UmbA/B/C. We are very pleased with these new data which provide a more accurate and nuanced model of Umb1. Furthermore, as described in the text section *Structure of the Umb1 particle* and referred to below in responses to Reviewers 3 and 4, we adopt an alternative and better supported rationale for defining the composition of the proteins within our Umb1 particle model.
- 2) We have replaced the data in Figure 4f,g with new data utilizing a more definitive control. In the original submission, we used supernatant deriving from *S. coelicolor* $\Delta umbC1-3$ as a negative control to support our contention that UmbC2 inhibits hyphal growth of *S. coelicolor*. We have repeated the entirety of this experiment utilizing the more specific control, $\Delta umbC2$, and present these new data in the revised manuscript. As expected, $\Delta umbC2$, like our original control, fails to inhibit hyphal growth of *S. griseus*.
- 3) In our original manuscript, we had not yet genetically complemented our Umb2-dependent phenotypes. By placing *umbC2* under the control of its native promoter at a neutral site on the *S. coelicolor* chromosome, we achieved full genetic complementation of $\Delta umbC2$ and present these data in *new* Fig. 4d of the revised manuscript.
- 4) In our original manuscript, we speculated based on our experimental structure and AlphaFold-based predictions that ALF6 of UmbC1 does not bind UmbB1. Furthermore, we proposed the positions of divergent amino acids in ALF6 that would preclude its binding to UmbB1 relative to UmbB1-binding ALF repeats (e.g. ALF2, etc.). In the revised manuscript, we validate these predictions in IP experiments that demonstrate ALF6 does not associate with UmbB1 and that a single ALF6-specific substitution within ALF2 abrogates its binding to UmbB1 (*new* Fig. 3f).
- 5) We identified a few IP experiments in the original submission that we felt the conclusiveness of would be strengthened by improved blot quality and/or loading normalization. Thus, we repeated these and present results with improved clarity in the revised manuscript (*revised* Fig 2c,f, j,k).
- 6) Our manuscript title has been modified to read "Umbrella toxin particles produced by Streptomyces block growth of vegetative hyphae in competing species" and in multiple locations throughout the text we have changed our wording when describing the effects of Umb particles on Streptomyces. The title change and these additional changes were

implemented to improve precision and accuracy with regard to describing the various life stages and growth modes of streptomycetes.

#####

Referee #1:

The model is probably correct and the 3D map file (at least the sharpened map) presented looks like a reasonable low-resolution map. However, the way the cryo-EM data is presented makes the map look better than it is. There is enough missing information to raise suspicion in a skeptical reader, which is not in the authors' best interests.

We agree and regret any missing information or confusion in our original presentation of the structural component of our manuscript. We address each issue pertaining to the structure individually below.

We also wish to highlight in this response that our revised manuscript brings a significant improvement to the quality of our structure. Using recently developed AI-based methods for particle identification (Topaz, Bepler *et al.* 2019), we roughly tripled the number of Umb1 particles incorporated into our dataset (Extended Data Table 1). While this does not negate the inherent challenges to structural characterization of Umb1 particles due to their marked compositional heterogeneity, it did significantly improve overall map quality and resolution from 5.1 Å to 4.3 Å and allowed us to obtain a 4.0Å resolution of a spoke using local refinement, thereby supporting model building and providing a blueprint of the interactions between UmbA/B/C. We are very pleased with these new data and we invite the reviewer to inspect the map files included with our resubmission. Our maps allowed side chain inclusion in certain locations and atoms were not modeled into areas of the map with weak density (e.g. lectin domains of spokes 2, 4 and 7).

I believe Fig. 3c shows a map calculated from the model, not the experimental map. That needs to be made clear.

There appears to be some confusion here. In the original Fig. 3c, the surface displayed was a chimera-generated molecular surface, and not a map derived from a low-pass filtered version of the model of the Umb1 particle, as we believe the reviewer is describing.

Fig. 3 has been entirely re-rendered with the new data, and we have clarified details pertaining to our presentation of the Umb1 particle in the substantially revised Fig. 3 legend.

The map shown in Fig. 3d appears to have been “cleaned up” – more than just using the “hide-dust” feature of Chimera. With the experimental map provided it is not possible for me to find a threshold where the stalk helices and the domains at the top of the structure are shown clearly at the same time. Has the map in Fig. 3d been shown at multiple thresholds? Is it a composite map? The details of what is shown need to be clear.

The map in Fig. 3d of our original submission was an experimental map that was segmented based on the different chains of the model. This explains why the reviewer was unable to generate a view from the overall map with a single threshold that precisely recapitulates Fig. 3d. The experimental map shown in Extended Data Fig. 7 was presented at a single threshold.

In *revised* Fig. 3 of our revised manuscript, we provide two maps deriving from our Umb1 particle data collection and processing: the transparent unsharpened cryo-EM map and the sharpened cryo-EM map. We selected these maps in order to highlight the overall ultrastructure of the Umb1 particle evident in our data (unsharpened cryo-EM map) and the quality of data we used to build our model (sharpened cryo-EM map). We have revised the Fig. 3 legend to clearly describe the maps presented. Finally, related to this reviewer comment, in *revised* Extended Data Figure 6e,f we provide our model within the cryo-EM map of the Umb1 particle, as well as insets that illustrate its relative quality at topologically distinct regions in the structure.

The methods section for the manuscript describes 2D classification as part of the image analysis process but 2D classes from the cryo-EM images are not shown anywhere (2D classes in Fig. S6C are from negative stain). Please show these 2D class averages from cryo-EM.

This was an oversight. We have added 2D class averages of our cryo-EM data to the revised manuscript (*new* Supplementary Fig. 2d).

The only FSC curves shown are for map-vs-model. Why is a standard halfmap-vs-halfmap FSC curve not shown? This curve is necessary to believe that the map is valid.

The requested halfmap-vs-halfmap FSC curve was provided in Fig. 7a of our original submission; however, we used the “gold-standard” nomenclature convention to describe this. Our revised manuscript includes the halfmap-vs-halfmap FSC curve (*revised* Extended Data Fig. 6a), a map-vs-model FSC curve (*revised* Extended Data Fig. 6b), and halfmap-vs-halfmap FSC curves for each local refinement (*new* Supplementary Fig. 4).

It is hard to imagine that a particle like this one would show isotropic orientations on the cryo-EM grid. Please provide an orientation distribution plot, 3D FSC plot. It is also hard to imagine that resolution in the map is uninform. Please provide a map of local resolution.

This is a great suggestion and we have added each of these to the revised manuscript. While we did observe moderate bias in the distribution of orientations, we observe particles in all orientations and this did not prohibit our reconstructing the full map. This was achieved by combining data from lacey grids with a continuous carbon support and from UltraFoil grids. The revised manuscript includes 3D FSC plots, orientation distribution plots and local resolution maps for the complete map and each local-refined map (*revised* Extended Data Fig. 6 and *new* Supplementary Fig. 4). At the suggestion of Reviewer #3, we have also added a cryo-EM workflow figure (*new* Supplementary Fig. 3).

I think the structure is probably valid but the details of the structural analysis should be shown with more transparency.

We thank the reviewer for their constructive remarks and we hope that the additional structure validation and methodology figures we describe above, in conjunction with clarified methods, provide them every confidence in the validity of our map and model.

#####

Referee #2 (Remarks to the Author):

Congratulations to Dr. Mougous and colleagues on a fantastic set of discoveries. This manuscript reports the discovery of an entirely novel family of large protein toxins in Actinobacteria. An depth look at examples of these toxins from *Streptomyces coelicolor* shows that the toxins rapidly arrest the vegetative mycelial growth of competitors, revealing a fascinating new mechanism of competition within the complex environment of the soil. Further, the toxins are assembled with a remarkable set of structural building blocks, revealing architectures heretofore unobserved in nature. This paper is paradigm shifting and opens up a new field with a wealth of new possibilities for understanding microbial soil ecology, novel antibiotic strategies, and principles of evolution.

The investigators have performed a tour-de-force study with an integrated and rigorous application of genomics, alpha-fold, cryo-EM, protein biochemistry, and microbiology . The data quality and data presentation are outstanding and allow for a robust set of conclusions. While the need for statistical analyses was limited, it was applied appropriately when needed.

I do not have any substantive suggestions for further improving this manuscript. It was masterfully written with an amazing attention to detail and clarity.

We are greatly appreciative and humbled by the kind words of the reviewer. Furthermore, we were pleased to find that the reviewer so accurately summarized what we too believe to be the key findings of the study.

It is our hope that the reviewer finds agrees that our minor revisions further improve the presentation of our study.

Minor points:

Line 338, typo in spelling of altruistic.

This typo has been corrected.

Line 351, in the list of open questions, the authors could also mention the presence of immunity proteins in some strains as this was mentioned early but not readdressed.

This is a great suggestion and we have added "... what role to immunity proteins play in protection against cis and trans intoxication, ..."

Line 355, typo in spelling of diphtheria

We have corrected our typos in the spelling of *Corynebacterium diphtheriae*.

#####

Referee #3 (Remarks to the Author):

This paper reveals the existence of a novel class of secreted antibacterial protein complexes made by Streptomyces, termed Umbrella (Umb) Toxin Particles. The particles consist of a long coiled-coil capped by a ring of toxin repeats, bearing 5 spokes with lectin tips. Bioinformatic analysis shows that these toxin particles are encoded not just within the Streptomycetaceae but also in 6 other actinobacterial orders. The potential diversity of these particles is striking. They incorporate lectin domains via promiscuous UmbA binding, and these lectins are highly variable in nature. In addition, at least 77 types of predicted toxin domains are associated with the UmbC proteins identified bioinformatically, and these toxins have very diverse predicted cellular targets (pore-formers, rRNases, mRNases, DNA deaminases, etc, etc). The authors screen a panel of 140 bacteria, biased toward Streptomyces spp., to look for species sensitive to Umb toxins. Through this route, they identify 2 target bacteria. They show that Umb particle-enriched supernatant from *S. coelicolor* selectively inhibited the growth of 2 Streptomyces species, *S. griseus* and *S. ambofaciens*, but not the growth of the other Streptomyces species included in the screen. In the case of *S. griseus*, time-lapse imaging showed that particle-enriched supernatant caused immediate growth arrest in actively growing vegetative mycelium but did not affect germination. Genetic inactivation of *S. coelicolor* umbC2, but not umbC1 or umbC3, produced particle-enriched supernatant that lacked the ability to inhibit *S. griseus*, consistent with Umb2 particles being responsible for the toxicity. The authors propose that the variable UmbA lectin domains mediate selective target cell binding, but do not address this question experimentally.

The Umbrella Toxin Particles described here represent an important discovery. Unsurprisingly, this work raises many additional biological and mechanistic questions, most of them enumerated by the authors themselves in the closing paragraph of the Discussion. How do the toxins with cytoplasmic targets cross the cell envelope, do the UmbA lectin domains determine target cell recognition, and what are the receptors on the target cells? Nevertheless, given this is the first description of these novel toxin particles, I think the content of the current paper represents a sufficient advance to be considered further for publication. However, there are key issues concerning the structural biology, for which there is a significant lack of detail that prevents proper scrutiny. There is also a troubling blurring of AlphaFold prediction into cryo-EM 'structure'. It is important that these issues are resolved.

BIOLOGY

1. The competition assays were done in liquid. Is there a similar inhibitory effect on vegetative growth on solid medium?

In preliminary experiments, we did not observe similar effects on solid medium. However, it is worth noting that we did not investigate this microscopically, which is conceivable where one

might anticipate such an effect to be discernible. This is especially the case for a large protein complex such as Umb particles, which would likely not diffuse far from their site of production.

2. Why was the screen of 140 potential target bacteria done with an undefined particle-enriched supernatant, rather than with multiple purified and defined Umb particle preparations?

This is a great question. In fact, since we had purified Umb1 particles in-hand, we initially conducted screening of potential targets with this material. However, we failed to obtain hits and realized that because 1) we could not be sure of the activity of our purified, tagged protein (in the absence of a positive control) and 2) we were only able to obtain screening-level quantities of purified Umb1, not other Umb particles produced by *S. coelicolor*, we would increase our odds of identifying a Umb target if we screened using Umb-enriched culture supernatant. Indeed, our finding that *S. griseus* is a target of Umb2, and not Umb1, reinforced the utility of this approach. In future work, we do anticipate describing targets of Umb1.

3. If the culture supernatant from *S. coelicolor* is treated with a protease, does this abolish the inhibitory effect on the growth of *S. griseus*?

We have not attempted precisely the experiment described by the reviewer, but we have shown that heat treatment of *S. coelicolor* Umb-enriched supernatant abrogates Umb2-dependent *S. griseus* growth inhibitory activity. We did not include these data in the manuscript, but provide them in Reviewer Response Fig. 1.

4. In the Introduction, in the context of known protein toxin systems in *Streptomyces*, it might be appropriate to reference the recently described cytoplasmic contractile injection systems (CIS) and their associated protein toxins, which appear to be involved in developmentally regulated cell death. Casu *et al.* (2023) *Nat Microbiol.* 8:711-726. Vladimirov *et al.* (2023) *Nat Commun.* 14:1469.

The reviewer raises a great point and one that we carefully considered when drafting our initial submission. However, we elected not to include a discussion of this system in our introduction because the eCIS of *S. coelicolor* are not polymorphic toxin systems to our knowledge and they are not known to mediate interspecies antagonism. Unfortunately, given space constraints, we feel that the length of explanation needed to include a reference to these systems does not justify their inclusion. Hopefully in future publications we will have an opportunity to discuss eCIS in the context of proteinacious toxins produced by *S. coelicolor*.

STRUCTURAL WORK

5. Line 180: 'Relative abundance in mass spectrometry data' is insufficient evidence for assignment of subunits in a cryo-EM reconstruction. Were there features in the cryo-EM map

that supported the assignment? If so, these need to be presented in the Extended data so that the level of confidence can be assessed by readers. Likewise, were the assigned positions of UmbA4, A5 and A6 clearly identifiable from the density, or assigned arbitrarily? If the latter, there is a significant risk of implying the confidence in the presented model is greater than it is.

We appreciate this point raised by this reviewer and by Reviewer 4. Owing to the UmbA heterogeneity within Umb particles and the averaging of single particles involved in cryo-EM, one cannot specify a particular UmbA protein in any single spoke of the Umb1 particle. We regret not making this sufficiently clear and acknowledge that our assignment of the four UmbA proteins to the five spokes of the Umb1 particle was arbitrary.

This presents an interesting problem and one that we have given a lot of thought to since our our initial submission. In the revised manuscript, we thoroughly detail the unique structural challenge presented by umbrella particles and carefully guide the reader through the solution to this challenge that we arrived at, which is to model the UmbA1 protein into each spoke. A thorough description of our rationale is found on lines 180-197 of the revised manuscript, and we hope the reviewer finds it compelling.

We also wish to highlight in this response that our revised manuscript brings a significant improvement to the quality of our structure. Using recently developed AI-based methods for particle identification (Topaz, Bepler *et al.* 2019), we roughly tripled the number of Umb1 particles incorporated into our dataset (Extended Data Table 1). While this does not negate the aforementioned inherent challenge to structural characterization of Umb1 particles due to their marked compositional heterogeneity, it did significantly improve overall map quality and resolution from 5.1 Å to 4.3 Å and allowed us to obtain a 4.0Å resolution of a spoke using local refinement, thereby supporting model building and providing a blueprint of the interactions between UmbA/B/C. We are very pleased with these new data and we invite the reviewer to inspect the map files included with our resubmission. Our maps allowed side chain inclusion in certain locations and atoms were not modeled into areas of the map with weak density (e.g. lectin domains of spokes 2, 4 and 7).

6. Line 170: The reason for the limited resolution requires clarification. ‘Protein aggregation’ presumably implies few particle images were usable as the majority were overlapping? ‘Heterogeneity’ could imply conformational or compositional variation between particle images. Rather than only being a limit to attainable resolution, this could hold important information on natural variation in the sample composition and/or possible dynamics associated with molecular mechanism.

We thank the reviewer for raising this matter of potential confusion in our manuscript. We used the term heterogeneity to refer to the UmbA composition within each spoke (see preceding response), and we understand how this term applied to the whole particle could generate confusion. We have therefore modified and elaborated upon our description of the limitations of cryo-EM as applied specifically to the Umb1 particle (lines 180-197).

7. There is a significant lack of detail regarding the cryo-EM data processing workflow and the following omissions need to be dealt with to allow reviewers to make a full assessment of the work. The number of particles (initial and final) is not presented in either the Methods or Extended Data figure legend. The classification of particle images leading the final reconstruction should be presented in the typical format of a ‘tree’ in the Extended Data where class features are indicated and selected subsets identified. Other pieces of information are also not provided that are standard practice are (i) reconstruction colored by local resolution, (ii) particle orientation distribution, (iii) representative 2D class averages, (iv) three-dimensional FSC to assess reconstruction quality (e.g. <https://3dfsc.salk.edu/upload/>). This information is crucial for Reviewers to assess whether an improvement on the modest resolution reported is achievable.

The lack of detail pertaining to our cryo-EM methods was noted by other reviewers as well. In response to the comment above and the cumulative requests of other reviewers, we have added the following to the revised manuscript: 1) 3D FSC plots (*new* Supplementary Fig. 4 and *revised* Extended Data Fig. 6a), 2) orientation distribution plots (*revised* Extended Data Fig. 6c and *new* Supplementary Fig. 4), 3) local resolution maps (*revised* Extended Data Fig. 6d, and *new* Supplementary Fig. 4), 4) a cryo-EM workflow tree (*new* Supplementary Fig. 3), and 5) 2D cryo-EM class averages (*new* Supplementary Fig. 2d).

8. The authors should consider a plot of reconstruction resolution for a series of given number of particles processed. This would indicate the number of additional particle images expected to be needed to reach a resolution sufficient for confident structural model construction (generally considered to be better than ~4 Angstrom resolution). Can the authors provide an estimate of how much additional data might be needed to achieve this with the current grid preparation conditions?

Thank you for this suggestion. As explained above, we have reprocessed the data using a new workflow which enabled to improve overall map quality and resolution from 5.1 Å to 4.3 Å and allowed us to obtain a 4.0Å resolution of a single spoke using local refinement. We encourage the reviewer to inspect the locally refined map and model, which is of very good quality for a nominal resolution of 4.0Å and provides a blueprint of the interactions between UmbA/B/C. The peripheral lectin domains of the UmbA proteins remain weakly defined due to 1) flexibility relative to the rest of the particle and 2) owing to their sequence divergence relative to the UmbA trypsin domains (recall the UmbA proteins are effectively an average, see description within “Structure of the Umb1 particle”).

9. The authors should also consider focused refinement, 3D variability analysis or similar tools for analysis of heterogeneous data. It is likely that a subset of the reconstruction could be improved beyond the reported resolution with such targeted data processing strategies. Can the authors confirm these were attempted and give the results?

Please see response to point 8 above. We have now performed local refinements focused on each spoke of the Umb1 particle encompassing the ALF repeat, UmbB1, and UmbA1. All maps are provided with this submission and local map resolutions are reported in *revised* Extended Data Table 1 and in *new* Supplementary Figure 4.

MINOR COMMENTS

Line 168: ‘structure’ should be ‘reconstruction’ or ‘map’, as these are the experimental data with a resolution metric.

This is very true, and we have modified our language describing the structure to avoid misrepresentating our findings.

Line 271. ‘Multiple species’ feels like an exaggeration for *S. griseus* + *S. ambofaciens*.

We have changed this to more accurately state “at least two species.” Given the relatively limited number of *Streptomyces* species included in our screen, and that the one we characterized further is intoxicated by only one of the three Umb toxins produced by *S. coelicolor*, it seems likely that more targets will be uncovered by additional screening.

Line 280-281. Is Fig 5b the correct figure to cite here?

This was a mistake, that has been corrected.

Line 352-353. “...mounted universally...” meaning not clear.

This has been changed to “universally employed”

TYPOS.

Throughout. Either “*Streptomyces*” (Italics, capital 'S') or “streptomycetes” (no italics, lower case 'S')

This has been corrected throughout.

Line 355. Typo. *Corynebacterium dipHtheriaE*

This typo has been corrected.

#####

Referee #4 (Remarks to the Author):

The manuscript by Zhao and co-authors describes an original, exciting and broadly-significant study identifying and characterising a new class of polymorphic toxins which occur widely in Actinobacteria and which are included within large multiprotein ‘umbrella’ particles unrelated to any previously-described type of anti-bacterial toxin delivery system. The authors show that these toxins block mycelial growth of competitor species and incorporate diverse anti-bacterial toxin domains, many previously unstudied. Additionally, the so-called umbrella toxins are able

to direct their own entry into susceptible target bacteria, perhaps using a modular lectin-like recognition of specific surface carbohydrates, opening up the possibility of future biotechnological or therapeutic utilisation. As would be expected following the discovery of a new type of toxin delivery system, many intriguing questions remain about how the particles are assembled, interact with target cells and deliver their toxin domain to the interior of the targeted cell. These are noted by the authors and will no doubt inspire future research spanning many areas to understand the mechanism and ecological significance of this system. The research has been conducted and presented to a high standard and an extensive body of data is included. Nevertheless, the authors should address one major concern and a number of minor comments, issues and suggestions as detailed below.

Major issue:

There is one major issue that the authors should address. The authors say that ‘based on their relative abundance in our mass spectrometry data’ they modelled UmbA1 into two of the five spokes in the cryoEM structure of the particle and UmbA4, UmbA5 and UmbA6 each into one of the spokes. This is not a convincing rationale. Spectral counting mass spectrometry cannot be used to determine the amounts of different proteins relative to each other in a quantitative manner. Even if the spectral counting data were to be considered in a semi-quantitative way, they still do not support the conclusion. In the UmbC1 IP, A1 gives 53 counts (with 6 in the control), A4 gives 49, A5 gives 40 and A6 gives 17, so not most consistent with 2:1:1:1 as proposed. In the UmbA1 IP, A1 gives 145 counts, A4 gives 51.5, A5 gives 49.5 and A6 gives 26. However, here A1 is the bait protein which would be expected to give a higher and non-stoichiometric signal since non-complexed protein in the cell will also be IP’d; whilst the other three, similar to the previous experiment, give an apparent stoichiometry of 2:2:1, not 1:1:1. The authors should provide further evidence to support their conclusion that the particles contain UmbA1, A4, A5 and A6 at a ratio of 2:1:1:1, and that two of the five spokes of the Umb particle contain A1 whilst the other three contain A4, A5 and A6 as assigned in the structure. Can the authors be certain that individual Umb particles do actually contain all four UmbA proteins simultaneously?

We appreciate this point raised by this reviewer and by Reviewer 3. Owing to the UmbA heterogeneity within Umb particles and the averaging of single particles involved in cryo-EM, one cannot specify a particular UmbA protein in any single spoke of the Umb1 particle. We regret not making this sufficiently clear and acknowledge that our assignment of the four UmbA proteins to the five spokes of the Umb1 particle was arbitrary.

This presents an interesting problem and one that we have given a lot of thought to since our our initial submission. In the revised manuscript, we thoroughly detail the unique structural challenge presented by umbrella particles and carefully guide the reader through the solution to this challenge that we arrived at, which is to model the UmbA1 protein into each spoke. A thorough description of our rationale is found on lines 180-197 of the revised manuscript, and we hope the reviewer finds it compelling.

We also wish to highlight in this response that our revised manuscript brings a significant improvement to the quality of our structure. Using recently developed AI-based methods for

particle identification (Topaz, Bepler *et al.* 2019), we roughly tripled the number of Umb1 particles incorporated into our dataset (Extended Data Table 1). While this does not negate the aforementioned inherent challenge to structural characterization of Umb1 particles due to their marked compositional heterogeneity, it did significantly improve overall map quality and resolution from 5.1 Å to 4.3 Å and allowed us to obtain a 4.0Å resolution of a spoke using local refinement, thereby supporting model building and providing a blueprint of the interactions between UmbA/B/C. We are very pleased with these new data and we invite the reviewer to inspect the map files included with our resubmission. Our maps allowed side chain inclusion in certain locations and atoms were not modeled into areas of the map with weak density (e.g. lectin domains of spokes 2, 4 and 7).

Other minor comments / issues to address:

(1) Extended Data Figure 6. In part a, left hand panel, the point corresponding to UmbC1 appears to be missing (or incorrectly placed further down the panel); according to the spectral counting data in Supplementary Table 3, UmbC1 should be present at around $\log_2 6$ (5.99), whereas no protein was detected with enrichment $\log_2 \sim 1.5$ as is currently plotted for the middle point.

We thank the reviewer for noting this issue. The point representing UmbC1 did in fact get lost from the graph during figure formatting and has been added back in the revised manuscript. In making this correction, we also discovered that we had used an early version of our filtering criteria, from before all replicates had been completed, in generating the graphs shown in Fig. 1C as well as Extended Data Fig. 6. We have corrected these in the revised manuscript, and the figures now accurately reflect the complete dataset shown in Supplementary Table 3. Importantly, the small changes we made (removing a handful of non-Umb proteins that failed to meet the final filtering criteria from the enrichment plots) do not alter the conclusions drawn from these experiments, and in fact serve to highlight the specificity of the enrichment for Umb proteins in the precipitations.

(2) Fig. 3a and Extended Data Fig. 6b. It is not totally clear what the difference is between these two panels/results – does Fig. 3a show the result of the UmbA1-V IP rather than the UmbA1-8xHis IP? This should be clarified in the legend to Fig. 3a. Were the UmbA1-V IPs also done on culture supernatant as for the UmbA1-8xHis IPs? This should be noted (UmbA1-V IPs are not specifically described in the Methods currently).

The reviewer is correct that Fig. 3a actually shows the result of the UmbA1-V IP, and the figure is mislabeled. The mass spectrometry analysis presented in Extended Data Fig. 6a (now renamed Extended Data Fig. 5a) derives from this sample. We opted to show the SDS-PAGE analysis of this sample in the main body, as we could include the directly comparable control sample. Extended Data Fig. 5b depicts the analysis of the UmbA1-8xHis IP, which was the material used for structural analysis. This sample underwent additional purification steps, and thus no analogously processed sample is appropriate to show as a control. We regret the confusion our mistake and lack of detail caused here and have now clarified what is shown in the respective figure legends.

(3) UmbD does not appear to be isolated from the UmbA1-8xHis and UmbA1-V IPs (unlike the initial UmbC1-V IPs). Does this reflect the fact that the presumed immunity protein is not secreted with the Umb particle, and does this have any implications for the mechanism of self-resistance of the producing cell prior to secretion?

We believe that the lack of UmbD1 in the UmbA1-8xHis and UmbA1-V IPs is related to cleavage of the toxin domain, mediated by the HINT domain found near the C-terminus of UmbC1. In the UmbC1-V IP experiments, the C-terminal placement of the epitope tag ensures that only particles precipitated are those which maintain the toxin domain, and thus are more likely to also contain the immunity protein.

(4) Starting on Page 10, the authors refer to screening for ‘Umb targets’ and ‘toxin targets’ but only later explicitly state that by ‘target’ they mean ‘susceptible organism’ or ‘target organism’. Given that the ‘target’ of a toxin is typically used and interpreted to mean the direct molecular target or substrate of a toxin, suggest to clarify their use of the word target as ‘susceptible target organism’ or equivalent on first usage in the text and the legend to Figure 4.

We have made this change as suggested.

(5) Line 208/209 and legend to Figure 4a. The authors state that heterologous expression of the C-terminal domains of the UmbC proteins leads to a drop in ‘bacterial viability’, whilst the figure depicts transformation efficiency of the plasmid rather than e.g. viable count. This is easily resolved and clarified if the authors note in the text or the legend that the transformation was done under conditions where expression of the toxin was induced and so number of viable transformants represents a read-out of toxicity.

In the legend to Figure 4a of the original submission, we stated that panel 4a represents, “Transformation efficiency in *Staphylococcus aureus* of plasmids **expressing** the indicated UmbC toxin domains relative to a vector control.” (emphasis added) We have been unable to find another succinct way to indicate that the transformations were performed under toxin-inducing conditions.

(6) Figure 4e. The y-axis title says ‘log Sc/Sg’ but the axis labels are not the log transformed numbers.

This has been corrected in the revised manuscript.

(7) Figure 5a. Could be useful to provide an indication/key for how the size of the boxes at the branch tips relates to the number of genomes included in that tip.

We appreciate this suggestion but could not find a way to readily incorporate this information into the figure, given space constraints. We have instead included a note in the legend to indicate the significance of box sizes.

(8) Figure 5b. The legend could indicate more clearly that these are predicted functions of the toxin domains, particularly in the case of 4TM_tox where the label ‘pore-forming’ is not yet

supported by experimental characterisation of these toxins or any homologous proteins.

We have made this edit as suggested.

(9) Line 291. Should be Fig. 5b

This has been corrected.

(10) Line 300. Should be Fig. 5c,d

This has been corrected.

(11) Methods section starting line 670. Multiple instances of ‘u’ instead of Greek Mu for ‘micro’

This has been corrected throughout the methods section.

(12) The authors mention a custom script (Line 752) which could be made available through GitHub.

Mention of a custom script here was actually a mistake, reflecting an early version of the analysis that was conducted. In the final comprehensive identification of conserved residues of toxin domains identified in this study, the program Chroma was employed, and this is now clarified in the text.

(13) For the microscopy experiment (Fig 4f, g and Extended Data Fig. 11), it is not clear if the experiment was repeated independently and on how many occasions.

The microscopy experiment was repeated independently a total of three times. This information is now included in the legend of Fig 4.

(14) The authors could comment on why there is not always a gene encoding an immunity (UmbD) protein associated with genes encoding UmbC toxins.

This is a question we currently do not know the answer to. There are a number of possibilities which we felt were too numerous to include a discussion of, given space restraints.

Reviewer Reports on the First Revision:

Referees' comments:

Referee #1 (Remarks to the Author):

The authors have address my concerns.

There a few remaining minor points, mostly related to use of language:

The caption for Figure S2 is "Full fields of view and 2D class averages of Umb1 particles from SEM and cryoEM". I think the authors use the acronym SEM to mean negative stain electron microscopy (without defining it as such), but the standard meaning of SEM is "scanning electron microscopy". These are not SEM images as most people would understand the acronym.

The text within the cut-and-paste FSC curves in Figure S4 is highly pixelated and unreadable.

Figure S4 and Figure ED6 use the phrase "gold standard threshold of 0.143". The phrase "gold standard FSC" refers to a "gold standard refinement" where the two half maps are kept separate during refinement. There is no "gold standard threshold".

The authors should confirm that the FSC curves in Figure S4 and ED6 were calculated with correction for the effects of masking.

Referee #3 (Remarks to the Author):

I am satisfied with all the changes in response - the presentation of the cryo-EM data now conforms to expectation.

The higher resolution (4.0 - 4.3 Å) in the revised MS is a significant increase from the initial submission and was achieved in a short space of time. This suggests that if the authors collected more EM data, they could readily achieve a further improvement in the structural model. Only ~4000 micrographs were collected, whereas a more typical dataset would be 10,000+. If this amount of data were analysed, the map would probably be <4Å resolution and the model would achieve high confidence. Because none of the conclusions in the manuscript depends on e.g., examining the positions of side chains, I think it would be unfair to insist, but in my eyes, it represents a missed opportunity to publish an optimal manuscript.

Referee #4 (Remarks to the Author):

The authors have substantially revised the manuscript, resulting in a more complete, polished and nuanced output. I still believe that this is an important and exciting study for the reasons detailed in my original review. The authors have addressed my concerns in the revised version.

I would like to make two minor suggestions based on the revisions:

(1) Lines 203-204 ' Subsequent IP experiments could not detect UmbB1-ALF6 binding, consistent with our structure'. New Figure 3f should be cited here to support the statement.

(2) The authors have clarified the nature of the IPs shown in Fig. 3a and Extended Data Fig. 6b. However the Methods still does not contain any information about how the UmbA1-V IP from culture supernatant was performed (only how it was performed from cell lysates). This information should be included.

Author Rebuttals to First Revision:

Reviewer comments in BLACK

Author responses in BLUE

We thank the reviewers for their careful reading of our revised manuscript. We have modified the manuscript to correct the errors and remove the points of potential confusion they have pointed out, as detailed below.

#####

Referee #1 (Remarks to the Author):

The authors have address my concerns.

There a few remaining minor points, mostly related to use of language:

The caption for Figure S2 is “Full fields of view and 2D class averages of Umb1 particles from SEM and cryoEM”. I think the authors use the acronym SEM to mean negative stain electron microscopy (without defining it as such), but the standard meaning of SEM is “scanning electron microscopy”. These are not SEM images as most people would understand the acronym.

We thank the reviewer for pointing out our typo here, which we have now corrected.

The text within the cut-and-paste FSC curves in Figure S4 is highly pixelated and unreadable.

We have increased the resolution of the text within the FSC curves in Figure S4 to address this problem.

Figure S4 and Figure ED6 use the phrase “gold standard threshold of 0.143”. The phrase “gold standard FSC” refers to a “gold standard refinement” where the two half maps are kept separate during refinement. There is no “gold standard threshold”.

These figure legends have been revised to indicate that the graphs show the FSC curve with a cutoff at 0.143.

The authors should confirm that the FSC curves in Figure S4 and ED6 were calculated with correction for the effects of masking.

We confirm that the FSC curves were corrected for the effects of soft masking by high-resolution noise substitution. This information has been added to the methods section of the revised manuscript.

Referee #3 (Remarks to the Author):

I am satisfied with all the changes in response - the presentation of the cryo-EM data now conforms to expectation.

The higher resolution (4.0 - 4.3 Å) in the revised MS is a significant increase from the initial submission and was achieved in a short space of time. This suggests that if the authors collected more EM data, they could readily achieve a further improvement in the structural model. Only ~4000 micrographs were collected, whereas a more typical dataset would be 10,000+. If this amount of data were analysed, the map would probably be <4Å resolution and the model would achieve high confidence. Because none of the conclusions in the manuscript depends on e.g., examining the positions of side chains, I think it would be unfair to insist, but in my eyes, it represents a missed opportunity to publish an optimal manuscript.

We appreciate this suggestion and regret any confusion our reviewer responses might have generated. The means by which we improved Umb1 particle resolution during revision was limited to improved particle picking within previously acquired micrographs, and not new data collection. In fact, we have found that purifying Umb1 particles amenable to cryoEM analysis is quite challenging and time-consuming, owing to their instability and tendency to aggregate.

Referee #4 (Remarks to the Author):

The authors have substantially revised the manuscript, resulting in a more complete, polished and nuanced output. I still believe that this is an important and exciting study for the reasons detailed in my original review. The authors have addressed my concerns in the revised version.

I would like to make two minor suggestions based on the revisions:

(1) Lines 203-204 ' Subsequent IP experiments could not detect UmbB1-ALF6 binding, consistent with our structure'. New Figure 3f should be cited here to support the statement.

We thank the review for pointing out our oversight in not citing the appropriate figure panel with this statement.

(2) The authors have clarified the nature of the IPs shown in Fig. 3a and Extended Data Fig. 6b. However the Methods still does not contain any information about how the UmbA1-V IP from culture supernatant was performed (only how it was performed from cell lysates). This information should be included.

We apologize for the additional confusion here. UmbA1-VSV-G, like UmbC1-VSV-G, UmbC2-VSV-G and UmbC3-VSV-G, was immunoprecipitated from a preparation of an *S. coelicolor* culture that included both cellular lysate and culture supernatant. We have clarified this in the figure legend for Fig. 3a, and in the methods section.